# IFNα induces CCR5 in CD4+ T cells of HIV patients causing pathogenic elevation
Hélène Le Buanec[1,11], Valérie Schiavon[1,11], Marine Merandet[1,11], Alexandre How-Kit[2], Hongshuo Song[3], David Bergerat [1], Céline Fombellida-Lopez[4], Armand Bensussan [1], Jean-David Bouaziz[1,5], Arsène Burny[6,7], Gilles Darcis[4], Mohammad M. Sajadi[3,7,8], Shyamasundaran Kottilil[3,7,8,9], Daniel Zagury[10,12] & Robert C. Gallo [3,8,12] ✉

## Abstract

**Background** Among people living with HIV, elite controllers (ECs) maintain an undetectable viral load, even without receiving anti-HIV therapy. In non-EC patients, this therapy leads to marked improvement, including in immune parameters, but unlike ECs, non-EC patients still require ongoing treatment and experience co-morbidities. In-depth, comprehensive immune analyses comparing EC and treated non-EC patients may reveal subtle, consistent differences. This comparison could clarify whether elevated circulating interferon-alpha (IFNα) promotes widespread immune cell alterations and persists post-therapy, furthering understanding of why non-EC patients continue to need treatment.

**Methods** Levels of IFNα in HIV-infected EC and treated non-EC patients were compared, along with blood immune cell subset distribution and phenotype, and functional capacities in some cases. In addition, we assessed mechanisms potentially associated with IFNα overload.

**Results** Treatment of non-EC patients results in restoration of IFNα control, followed by marked improvement in distribution numbers, phenotypic profiles of blood immune cells, and functional capacity. These changes still do not lead to EC status, however, and IFNα can induce these changes in normal immune cell counterparts in vitro. Hypothesizing that persistent alterations could arise from inalterable effects of IFNα at infection onset, we verified an IFNα-related mechanism. The protein induces the HIV coreceptor CCR5, boosting HIV infection and reducing the effects of anti-HIV therapies. EC patients may avoid elevated IFNα following on infection with a lower inoculum of HIV or because of some unidentified genetic factor.

**Conclusions** Early control of IFNα is essential for better prognosis of HIV-infected patients.

## Plain language summary

The treatment for HIV, known as antiretroviral therapy (ART), does not cure HIV but enables individuals to live longer, healthier lives. In this study, we compared immune responses between elite controllers (ECs), who control their HIV infection without any treatment, and ART-treated and untreated patients. We demonstrate that IFNα, a small protein crucial in controlling immune system, is excessively produced at the onset of HIV infection and at levels that persist, resulting in poor HIV control without therapy. We show a mechanism for lack of control of HIV by IFNα. While inhibiting HIV, IFNα also simultaneously increases the HIV co-receptor, CCR5, thereby facilitating virus entry into the target cell. This is avoided by ECs which we hypothesize is associated with a lower infectious inoculum of HIV.

Therapy for HIV infection has been based on targeting specific steps in HIV replication, leading to major clinical improvement and prognosis. However, some medical issues remain unresolved[1–6], and residual HIV reservoirs[7,8] lead to loss of virus control when therapy is interrupted[9,10]. The chief clinical goal currently is to attain a functional cure in which no further therapy is needed, mimicking elite controller (EC) status (0.5% of HIV-infected persons requiring no therapy for years or even decades)[11,12]. To this end and in keeping with what others have suggested[13], we think that new, relevant insights could be gained from a deeper understanding of the relevant pathogenesis. In this context, we previously examined all blood immune cell types at different stages of an HIV immune reaction, comparing untreated HIV-infected EC and non-EC patients[14]. We found that all immune cells were altered in non-EC patients, leading us to suggest the involvement of one or more mediators in addition to HIV, which directly targets only a few immune cell types. The main mediator we identified is elevated circulating pathogenic IFNα, which is active in the earliest days of infection in non-ECs but not in ECs[14], possibly because infection in ECs resulted from a relatively lower inoculum.

In this companion report, we pursue these studies further with a detailed analysis of all blood immune cells of HIV-infected non-ECs who

have received anti-retroviral therapy (ART) and a comparison of these results with those of untreated EC patients. If therapy is interrupted in treated non-EC patients, HIV viremia rebounds, demonstrating that anti-HIV therapy is not curative. We show here that therapy removes the bulk of HIV-specific immune cell alterations while controlling IFNα levels, as seen in ECs; however, some residual alterations persist, likely because of IFNα effects before therapy. Furthermore, we identify the cause of the failure in non-ECs of IFNα to contain HIV in the earliest days of infection, which leads to a vicious cycle of IFNα response to increasing HIV titres and ultimately to pathogenically high IFNα levels. In turn, non-EC patients sustain early immune cell damage that can persist in part even after therapeutic control of IFNα. Of note, as is usually the case, anti-HIV therapy in these patients was not begun in the earliest days of infection.

## Methods

### Human samples

A total of 73 Healthy donor (HD) samples were obtained through the Etablissement Français du Sang (EFS, Paris, France). 51 samples were used for serological analysis and 22 for phenotypic analysis. A total of 81 people living with HIV were recruited and subdivided into three groups: EC samples ($n = 18$) were obtained from the Natural Viral Suppressors (i.e., NVS) cohort (Baltimore), untreated non-EC patient (UP) samples (n = 36) were obtained from the National Institutes of Health (NIH; Bethesda $n = 19$) and from the Laboratoire de Référence SIDA (Liège $n = 17$), and treated patient (TP) samples ($n = 27$) from NIH (Bethesda $n = 19$) and from Laboratoire de Référence SIDA (Liège $n = 8$). Patient groups did not significantly differ in terms of age, gender, or disease status. All participants provided informed consent in accordance with protocols approved by the regional ethical research boards and the Declaration of Helsinki (H-29331 (USA) and 2020/418 (Belgium)). Clinical data are indicated in Supplementary Table 1a–c.

### Sample processing

Peripheral blood and serum were collected into appropriate tubes. Peripheral blood mononuclear cells (PBMCs) were isolated by density gradient centrifugation on Ficoll-Hypaque (Pharmacia, St Quentin en Yvelines, France). PBMCs were stored frozen in nitrogen liquid.

### Cell culture

CD4$^+$T cells were isolated from frozen PBMCs. All CD4$^+$T cells were positively selected with a CD4$^+$T cell isolation kit (Miltenyi Biotec, Bergisch-Gladbach, Germany), yielding CD4$^+$T cell populations at a purity of 96%–99%. Purified CD4$^+$T cells were stimulated as described previously[15] with 4 μg/mL plate-bound anti-human CD3 (OKT3) monoclonal antibody (mAb; eBioscience, San Diego, CA, USA) and 4 μg/mL soluble anti-human CD28 (CD28.2) mAb (Becton Dickinson) in the presence of recombinant human IL-2 (Proleukine, Chiron, Amsterdam, 100 U/mL) and recombinant human interferon alpha-2a (Roferon-A 0.01–100 ng/mL). After 4 days of culture, membrane CCR5 and CXCR4 expression was measured by flow cytometry on CD4$^+$T cells with an anti-CCR5 mAb (clone REA245 Miltenyi Biotec) and an anti-CXCR4 mAb (clone 12G5, BioLegend).

### Infectious molecular clones

The infectious molecular clone (IMC) of the CH058 transmitted/founder (T/F) virus (CCR5-tropic) was obtained from the NIH HIV Reagent Program (catalogue number: 11856). The CXCR4-tropic T/F virus 40700, which is unable to use CCR5, was identified in an acute CRF01_AE infection. The full-length genome of the 40700 T/F virus was determined using single-genome amplification, as previously described[16]. Detailed information about the 40700 IMC, including the full-length viral sequence, will be described in a separate publication.

### Viral stock preparation and titration

To generate viral stocks, 6 μg of IMC was transfected into 293 T cells in a T25 flask using the FuGENE6 transfection reagent (Promega). Six hours

post transfection, the culture medium containing the plasmids and the transfection reagent was completely replaced with 8 mL of fresh medium. The cells were cultured at 37 °C for 3 days. Culture supernatants were harvested at 72 hours post transfection, filtered, and stored at −80 °C until use. The infectious titres (TCID50) of the viral stocks were determined on TZM-bl cells.

### Determination of virus growth kinetics in PBMCs

Cryopreserved PBMCs from one HD were thawed and recovered overnight at 37 °C in RPMI1640 containing 10% FBS. The next day, the cells were stimulated with 4 μg/mL plate-bounded anti-CD3 (clone OKT3, eBioscience) and 4 μg/mL soluble anti-CD28 (clone CD28.2, eBioscience) in the presence of 100 U/mL IL-2 (Human IL-2 IS, Miltenyi Biotec) in a 96-well U-bottom plate at a density of $0.3 \times 10^6$ cells/well. To determine whether upregulation of CCR5 by IFNα could compromise the inhibitory effect of IFNα on HIV-1 replication, cells were pretreated with three different concentrations of IFNα (0.2 ng/mL, 1 ng/mL, 5 ng/mL) during the stimulation step. Cells stimulated in the absence of IFNα (non-pretreated) were used as a control. Four days after stimulation, cells were washed twice to remove the stimulating antibodies and infected by the CH058 or 40700 T/F virus at a multiplicity of infection of 0.05. At 4 h after infection at 37 °C, cells were washed three times with RPMI1640. The infected cells were cultured in a 48-well plate with 500 μL RPMI1640 containing 10% FBS, 100 U/mL IL-2, and a corresponding concentration of IFNα (0.2 ng/mL, 1 ng/mL, or 5 ng/mL). As a positive control, infected cells (non-pretreated) were cultured in the absence of IFNα to allow determination of the normal replication kinetics of the virus. Culture supernatants were harvested at 2, 6, 12, 24, and 48 h post infection, and virus replication was determined by measuring the p24 concentration in the supernatants.

### Antibody panels, staining, and flow cytometry analysis

Immunophenotypic studies were performed on frozen samples, using flow cytometry panels with up to 23 colours. Approximately $1 \times 10^6$ to $5 \times 10^6$ frozen PBMCs were used per patient per stain. PBMCs were thawed using RPMI supplemented with 20% FCS per standard protocol. Cells were resuspended in 45 μL FACS buffer (PBS supplemented with 0.5% BSA and 2 mM EDTA). To block Fc receptor binding, 5 μL of FcR Blocking Reagent (Miltenyi, Paris, France) was added to the cells for 10 min at 4 °C, following the manufacturer's instructions. Cells were washed once with FACS buffer before the staining. Staining was performed in several stages, depending on the number of antibodies. The first mAb cocktail for surface staining, prepared with brilliant ultraviolet, brilliant violet, and brilliant blue antibodies in Brilliant Stain Buffer Plus (BD Biosciences, Paris, France), was added to the cells and incubated for 30 min at 4 °C (see Supplementary Tables 2–4 for antibody panel information). Cells were washed once with FACS buffer before addition of the second mAb cocktail, which contained all other antibodies in FACS buffer for surface staining and viability staining, and incubated for 30 minutes at 4 °C. When needed, cells were fixed and permeabilized using the Foxp3 Staining Buffer Set (eBioscience, Paris, France) according to the manufacturer's protocol. Cells were then stained for intracellular targets with the third mAb cocktail in the kit's permeabilization buffer. After incubation, cells were washed with permeabilization buffer and FACS buffer. Before use within this panel, all antibodies were titrated individually to identify the concentration that provided the maximal brightness of the positive cell population and the lowest signal for the negative cell population. Cells were acquired on a Cytek Aurora flow cytometer. Data were analysed using FlowJo software (FlowJo, LLC). The gating strategy used to identify the immune cell subtypes and their respective subsets is depicted in Supplementary Fig. 1. Unsupervised analyses were performed using Cytobank software and R studio software.

### Composite cell phenotypic alteration score

We generated a cumulative phenotypic score for each T cell subset. The frequencies of the following markers were used to calculate the score: CD25$^-$, CD26$^-$, HLA-DR$^+$, CD38$^+$, CTLA-4$^+$, CD28$^-$, PD1$^+$, and CD39$^+$. This score

was calculated as the sum of the ratio of marker expression level to the average expression level of the corresponding marker in the HD.

## Cytokine quantification

Serum IFNα and IFNλ2 levels were determined using SIMOA cytokine assays (references 100860 and 101419, respectively). Van der Sluis et al. have reported that HIV infection of co-cultures of CD4 + T cells and plasmacytoid dendritic cells ([p]DCs) enhances expression of *IFNλ2* and not *IFNλ1* or *IFNλ3* mRNA, so we focused only on serum IFNλ2 levels[17].

## RNA extraction, reverse-transcription, and real-time qPCR analyses

Total cellular RNA was extracted using RNeasy RNA extraction columns (Qiagen) according to the manufacturer's instructions and stored at −80 °C. Total RNA (100 to 200 ng) was reversed transcribed using PrimeScript with gDNA Eraser (Takara Bio) with random hexamer primers. For real-time qPCR, cDNA and primers (CXCR4_F1: TCT-CGT-GGT-AGG-ACT-GTA; CXCR4_R1: CAC-TTT-GGG-CTT-TGG-TTA-T; CCR5_F1: GAC-TTA-GAA-CCA-GGC-GAG-AG; CCR5_R1: GCA-GTG-AGG-CTT-CTG-TCT-T) were added to Syto9 in 96-well plates and run on a LC480 thermocycler (Roche) with the following cycling conditions: 95 °C for 10 min; then 50 cycles of 95 °C for 30 s, 60 °C for 30 s, and 72 °C for 20 s; and a final step at 72 °C for 10 min. Dissociation curve analysis after the end of the PCR confirmed the presence of a single and specific product. Glyceraldehyde 3-phosphate dehydrogenase was used as endogenous control and as reference for relative quantification.

## Statistics and reproducibility

Comparisons between two unpaired groups were conducted with the Mann–Whitney U test. Comparisons between paired data were made using the Wilcoxon test. Multiple group comparisons were assessed with the Kruskal–Wallis test with Dunn's multiple comparison testing. Error bars on graphs represent interquartile ranges. Correlations were assessed by the nonparametric Spearman test. Analyses were performed with GraphPad Prism and R. A two-sided $p < 0.05$ was considered statistically significant (ns: not significant; * $p < 0.05$; ** $p < 0.01$; *** $p < 0.001$; **** $p < 0.0001$).

## Reporting summary

Further information on research design is available in the Nature Portfolio Reporting Summary linked to this article.

# Results

The aim of this study is to unravel how the anti-HIV effect of IFNα is impaired. As detailed in the methods, here we compare[1] IFNα and IFNλ2 serum concentrations[2], distribution of immune cell subsets, and[3] the frequency of cell markers associated with immune dysfunction in ECs and in untreated and treated non-EC patients, with HD samples as a control.

## Distinct blood immune cell profiles and serum IFNα levels in TPs compared with UPs, and HDs

We first used principal component analysis (PCA) to test whether the major immune cell subtype frequencies could distinguish TPs from UPs and HDs. The results show that UPs and HDs are clearly separated, while TPs are distributed between both groups (Fig. 1a). We next investigated immune features that may drive this TP immune pattern, showing no difference in CD3+T cell frequency among the groups (Fig. 1bI). Compared with HDs, however, UPs and TPs show some decrease in CD4+T cells (Fig. 1bII), with a concomitant increase in CD8+T cells (Fig. 1bIII). Furthermore, the frequency of γδ T cells is reduced in TPs compared with UPs (Fig. 1bIV). The frequency of DCs (lin-HLA-DR+) also is increased in TPs and in UPs compared with HD, though to a lesser degree (Fig. 1bV). Finally, the percentage of natural killer (NK) cells is similar between TPs and HDs but markedly reduced in UPs (Fig. 1bVI). The significant variation in the frequencies of these immune cell types across the three studied groups is summarized in a balloon plot (Fig. 1c).

We also evaluated serum IFNα and IFNλ levels among these three groups. Compared with UPs, TPs and HDs have highly significantly lower serum IFNα levels (Fig. 1dI). We observed no remarkable changes in IFNλ concentration among the three groups (Fig. 1dII). The variation in IFNs in paired samples collected before and after combination ART (cART) confirms that serum IFNα, but not IFNλ, disappears after treatment (Fig. 1dIII-dIV and Supplementary Table 5). Interestingly, IFNα level is positively correlated with IFNλ2 level in TPs (Fig. 1eII) as well in ECs[14], but not in UPs (Fig. 1eI).

## High IFNα level enhances in vitro expression of the HIV coreceptor CCR5

Beta chemokines are the natural ligands of CCR5, and as such are potent inhibitors of HIV[18,19]. Previously, we reported that elevated IFNα inhibits production of these HIV inhibitors[20]. We hypothesized that a greater and more immediate impact would occur if IFNα also directly increased the amount of the HIV coreceptor CCR5. Indeed, we show here that IFNα but not IFNλ2 enhances *CCR5* mRNA expression on stimulated CD4+T cells isolated from HDs (Fig. 2aI), resulting in higher levels of the CCR5 protein on CD4+T cells (Fig. 2aII). Interestingly, IFNα has no effect on CXCR4 expression in stimulated CD4+T cells at either the mRNA (Fig. 2aIII) or protein level (Fig. 2aIV). Similarly, IFNα induces a dose-dependent increase in CCR5 expression on stimulated CD4+T cells isolated from UPs, TPs, and ECs (Fig. 2b).

## Upregulation of CCR5 on CD4+T cells by IFNα impairs the IFNα anti-HIV effect

The CCR5 results suggest that the inhibitory effect of IFNα on CCR5 virus replication could be compromised beginning at the onset of infection and then persist throughout the innate phase of the immune reaction. To test this hypothesis, we infected HD PBMCs with CCR5 (R5) or CXCR4 (X4)-tropic transmitted/founder (T/F) HIV-1 and compared the inhibitory effect of IFNα in IFNα-pretreated and non-pretreated cells. For both the CCR5 and CXCR4 viruses, the exponential phase of virus replication begins at 24 h after infection (Fig. 2c). For the R5-tropic virus, the level of IFNα inhibition declines in IFNα-pretreated cells compared with non-pretreated cells. At 48 h post infection in non-pretreated cells, we found that virus production is inhibited by 39.3%, 51.8%, and 49.4%, respectively, at 0.2 ng/mL, 1 ng/mL, and 5 ng/mL of IFNα (Fig. 2cI). In comparison, in pretreated cells, we found that the same concentrations of IFNα inhibit virus production by only 16.3%, 22.9%, and 38.5%, respectively, at the same time point (Fig. 2cI). As expected, pretreatment with IFNα has no effect on inhibition of the X4-tropic virus. In contrast, pretreatment with 5 ng/mL IFNα enhances the inhibitory effect (Fig. 2cII). These observations indicate that upregulation of CCR5 by IFNα compromises its inhibition of virus replication for R5 HIV-1, setting the stage for greater damage by both HIV and a rising level of IFNα.

Considering that IFNα is correlated with HIV load, we assume that during the initial phase of infection, elevated IFNα, infected CD4+T cells, and HIV form a pathogenic vicious cycle from HIV infection to increased IFNα to increased viral load: HIV → IFNα → infected CD4+T cells → more HIV → more IFNα and so on.

## Persistence of some alterations in immune cell subtypes involved in the innate phase of the immune response in TPs

Other than IFNα, components of the innate phase of the immune response include specific cells, especially NK cells. We find differences among UPs, TPs, and HDs in the distributions of the three major NK cell subsets[21,22] (Fig. 3aI). The proportion of early NK cells in UPs and TPs is lower than in HDs. In addition, compared with HDs, terminal NK cells increase in UPs and to a lesser extent also in TPs (Fig. 3aII-aIII). We further find that NK cell subsets from UPs and TPs display distinct immune profiles associated with immune dysfunction. Mature NK cells from TPs still exhibit lower levels of Helios and natural cytotoxicity receptors (NCRs) and increased levels of

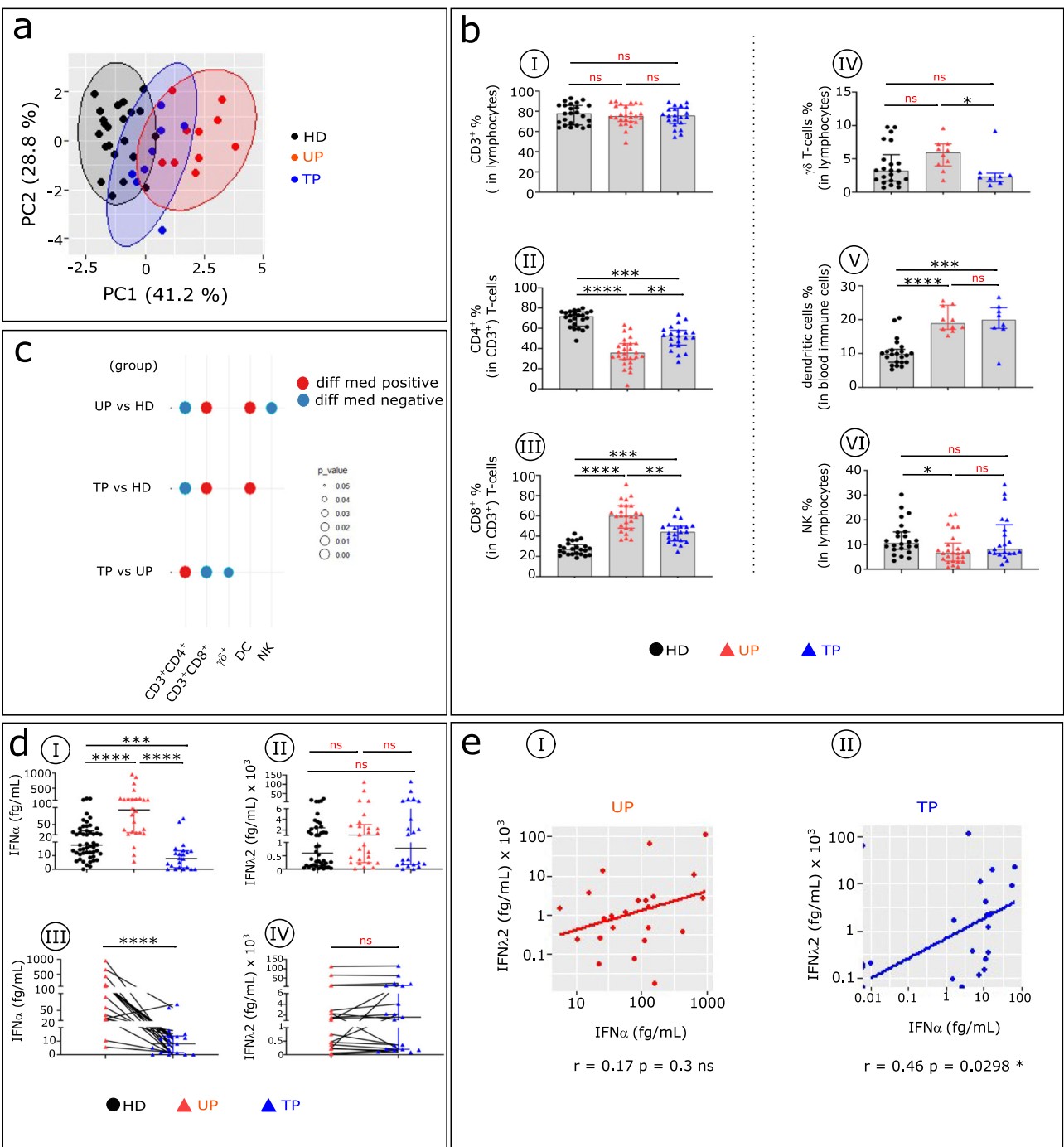

**Fig. 1 | Comparative analysis of major blood immune cell subsets and serum IFN concentrations in UPs, TPs, and HDs. a** PCA of the data is based on the proportion of different cell subpopulations (CD4+, CD8+, and TCR γδ T cells; NKs and DCs) evaluated by flow cytometry, as depicted in Supplementary Fig. 1. The first two principal components (PC1 and PC2), representing the greatest differences among individuals, are represented in a bi-plot. Each point represents one participant, colour-coded by group: HDs (black), UPs (red) and TPs (blue). Each group is outlined by an ellipse representing the 95% confidence interval of the sample groupings. **b** Histograms show distributions of indicated immune cell populations between HDs (black), UPs (red), and TPs (blue). Analysis was done in 24 HDs, 26 UPs, and 21 TPs for all populations except for γδ T cells and DCs (lin⁻ HLA-DR+) (22 HDs, 10 UPs, 8 TPs). **c** Balloon-plot summarizing the statistically significant

changes in the indicated immune cell populations between UPs and HDs, TPs and HDs, and TPs and UPs. The size of the circle indicates the *p* value. Red and blue respectively indicate increased and decreased frequencies of the immune cell populations. **d** Scatterplots of IFNα and IFNλ2 concentrations in serum from HDs (*n* = 51), UPs (*n* = 26), and TPs (*n* = 22). Levels of IFNα and IFNλ2 were detected by SIMOA in unpaired (D1, D2) and paired patients (D3, D4). **e** Scatterplot of the relationships between IFNα and IFNλ2 serum levels in the 22 paired UPs and TPs. Correlations were evaluated with Spearman's rank correlation test. Paired data were compared using the Wilcoxon test. Multiple group comparisons were assessed using the Kruskal–Wallis test with Dunn's multiple comparison testing. Values are medians and *p* values (**p* < 0.05, ***p* < 0.01, ****p* < 0.001, *****p* < 0.0001). Error bars on graphs represent interquartile ranges.

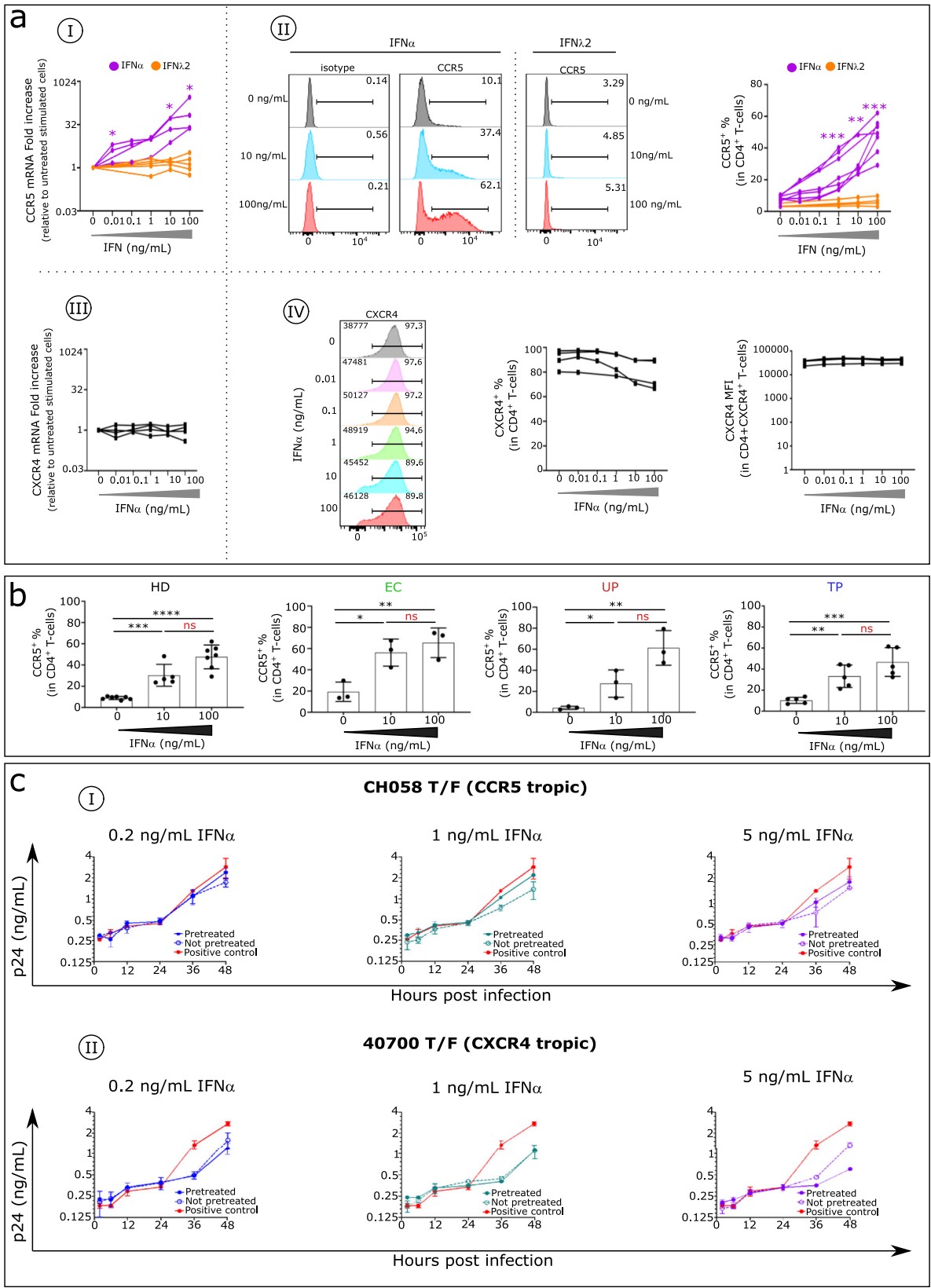

IFNα-induced CD38 and HLA-DR. In contrast, GrzB, CD26, CD39, inhibitory Killer-cell immunoglobulin-like receptors iKIR, and PD1 expression patterns on mature NKs from TPs and HDs are similar (Fig. 3aIV-aVI).

We next investigated the distribution of CD11c⁺ myeloid DC (mDC) and CD123⁺ pDC subsets in the three groups (Fig. 3b). Of note, the pDC frequency is reduced in both patient groups, although more so in UPs, and the mDC proportion also is increased in these groups, although more so in TPs (Fig. 3bII-bIII). The γδ T cells, another subset of innate immune response cells, also show functional signalling receptor expression abnormalities (Fig. 3c) with IFNα-induced HLA-DR, associated in some

**Fig. 2 | Elevated IFNα effect on HIV coreceptor CCR5 expression in human CD4⁺T cells and on release of circulating HIV by infected CD4⁺T cells. aI** *CCR5* mRNA expression levels, assessed by RT-qPCR, and (**aII**) CCR5 protein expression level, evaluated by flow cytometry in CD4⁺T cells stimulated with different concentrations of IFNα (mRNA analysis n = 4, protein analysis *n* = 7, purple) and IFNλ2 (mRNA analysis *n* = 5, protein analysis *n* = 4, orange). **aIII** *CXCR4* mRNA expression levels assessed by RT-qPCR and (**aIV**) CXCR4 protein expression level evaluated by flow cytometry in CD4⁺T cells stimulated with different concentrations of IFNα (mRNA analysis *n* = 4, protein analysis *n* = 4). **b** Histograms show CCR5 frequency in CD4⁺T cells stimulated with different IFNα concentrations in each studied group (HDs *n* = 7, ECs *n* = 3, UPs *n* = 3, and TPs *n* = 5). Multiple group comparisons were made using the Kruskal–Wallis test with Dunn's multiple comparison testing. Values are medians and p values (*$p < 0.05$, **$p < 0.01$, ***$p < 0.001$,

****$p < 0.0001$); ns: not significant. Error bars on graphs represent interquartile ranges. **c** PBMCs from a single HD were stimulated in the presence (pretreated) or absence (not pretreated) of IFNα for 4 days. The stimulated cells were infected by the CH058 T/F virus (CCR5 tropic) (**cI**) or the 40700 T/F virus (CXCR4-tropic) (**cII**). Upon infection, cells were cultured with the corresponding concentration of IFNα for 48 h. For the positive control, the infected cells (not pretreated) were cultured in the absence of IFNα for determination of the normal replication kinetics of the virus. The p24 concentration in the culture supernatants was measured at 2, 6, 12, 24, and 48 h post infection. The infections were performed in duplicate, and the error bar represents the standard deviation (SD). Experiments were performed in PBMCs from three HDs. The results of one representative experiment were shown. MFI: median fluorescence intensity.

cases with CD38. These alterations are still significantly more frequent in TPs than in HDs but less than in UPs.

Collectively, these results show that immune cells involved in the innate phase of the immune response still present functional abnormalities in TPs compared with HDs, but at a lower level than in UPs.

### Residual T cell phenotypic profile abnormalities in TPs

To compare immune T cell profiles among UPs, TPs and HDs, we characterized the T cell subtype frequency and phenotypes, as described in the material and methods and elsewhere[14]. UPs have an abnormal T cell distribution compared with HDs. The frequency of CCR7⁺ in CD3⁺, CD4⁺, and CD8⁺T cells (Fig. 4a) is decreased in UPs compared with HDs. This loss of T cells expressing CCR7 also is present in TPs, although to a lesser degree. Compared with HDs, UPs exhibit altered proportions of naïve, central memory (CM), effector memory (EM), and terminal differentiated EM CD45RA-expressing (TEMRA) cells among the CD4⁺ (Fig. 4bI–bII) and CD8⁺ (Fig. 4bIII-bIV) T cell subsets. TPs still show an abnormal differentiation profile but with normalization of their CM and EM CD8⁺ frequencies. In addition, UPs compared with HDs exhibit an altered phenotypic profile in T cell subsets (Fig. 4c). This difference can be attributed in part to high IFNα levels[14]. These phenotypic alterations are characterized by an abnormal level of activation/differentiation markers (CD38, HLA-DR, CD25, CD26, CD28) and inhibitory receptors (CD39, PD1, CTLA4) (Fig. 4c and Supplementary Fig. 2). In general, in contrast to UPs, TPs display immune profiles that are similar to those of HDs. However, some CD8⁺T cell phenotypes in TPs, such as HLA-DR⁺CM, CD38⁺HLA-DR⁺EM, remain significantly different from those of HDs. After treatment, partial reversion of T cell phenotype anomalies in all CD4⁺ and CD8⁺T cell subsets is observed, as evaluated using phenotypic alteration scores (see Methods) (Fig. 4cIII-4cIV). Interestingly, in UPs, expression levels of various markers associated with immune dysfunction in the CD4⁺ and CD8⁺ CM population are directly correlated (Supplementary Fig. 2aIV, 2bIV). This pattern is in keeping with the interpretation that these alterations are induced by a major mediator such as elevated IFNα. No such correlation is found in cells from TPs.

According to the phenotypic profile analysis of regulatory T cells (Tregs), the frequency of Tregs is greater in UPs compared with HDs (Fig. 5aI–aII). This increase in Treg frequency, however, is not seen in TPs. In contrast, we found that after treatment, patients still have a high frequency of Tregs lacking CD25 expression[15], with a higher frequency of this CD25^neg Treg variant in UPs (Fig. 5aIII-5aIV). This altered immune profile of memory Tregs observed in UPs is partially restored in TPs (Fig. 5aV).

CD8⁺T cell subsets from HIV-infected patients show quantitative and qualitative defects of differentiated CD8⁺ cytotoxic T lymphocytes (CTLs) (KIR⁻) and cytotoxic CD8⁺ suppressive T cells (CD8⁺supps; KIR⁺) in the TEMRA subset (Fig. 5b). In TPs compared with HDs, CTL is increased, with a concomitant reduction in CD8⁺supp (Fig. 5bI and bIII). These populations are similar in UPs and HDs. In addition, compared with HDs, UPs show a higher frequency of CTL and CD8⁺supp cells expressing various markers of activation, differentiation, and exhaustion, whereas TPs display a phenotypic pattern that is somewhat similar to that of HDs (Fig. 5bII, 5bIV).

Furthermore, IFNα-induced CD38 correlates with expression of inhibitory checkpoint proteins (CTLA-4 and PD1) in both CTL and CD8⁺supp in UPs, but not in TPs (i.e., after therapy) (Fig. 5bV).

In summary, the phenotypic anomalies associated with loss of immune cell function of conventional T cells (Tconvs) and Tregs are markedly reduced in TPs compared with UPs, although they show similar features between the two groups, and in large part these are proteins induced by IFNα. The CD8⁺ functionally linked phenotypic alterations also are of the same nature as those in CD4⁺T cells but intensified, and they are markedly reduced after therapy. These immune alterations in both Tconv and CD8⁺ cytotoxic T cells observed before therapy and still present after cART may account for the fragile health suggested by the comorbidities that persist or arise in TPs.

### TPs and ECs share immune profile features

We next analysed blood immune cell profiles in TPs compared with ECs (Fig. 6). viSNE and PCA analysis indicate that TPs and ECs display similar cell distributions that are distinct from those of UPs and HDs (Fig. 6aI-aII). Both TPs and ECs also have normal background levels of serum IFNα. One difference between them is in γδ T cells (Fig. 6aIII). There is no difference in the proportion of NK cell subsets between TPs and ECs, but TPs show a decrease in early NK cells compared with HDs (Fig. 6bI). Based on analysis of phenotypic abnormalities in mature NK cells, there is a higher frequency of CD38⁺ HLA-DR⁺ and PD1⁺ cells in TPs and ECs than in HDs (Fig. 6bII). Interestingly, TPs and ECs also share a similar percentage of DCs (Fig. 6cI-cII). One difference in the DC compartment is a significant increase in the mDC subset in TPs. Dysregulation of T cell homoeostasis, as observed in untreated HIV-1–infected patients[23], is maintained in TPs compared with ECs, although to a lesser degree than in the previous report. TPs have fewer CCR7⁺ CD4⁺T cells than HDs (Fig. 6dI). Regarding CD8⁺T cells, TPs and ECs have decreased frequencies of CCR7⁺T cells compared with HDs (Fig. 6dII). Furthermore, the few T cell phenotypic alterations seen in TPs and ECs are similar (Fig. 6eI, f I and Supplementary Fig. 3). Finally, analysis of Treg cells reveals that their frequencies are similar among the three groups (Fig. 6eII). Of note, TPs and ECs show an increase in the CD25^neg Treg variant compared with HDs (Fig. 6eIII). Within CD8⁺TEMRAs, TPs and ECs show an increase in CTLs (Fig. 6f II) with a concomitant decrease in CD8⁺supp cells (Fig. 6f III) compared with HDs. The anomalies observed in TPs and ECs are much less pronounced than in UPs but slightly higher than in HDs (Fig. 6f IV-f V and Supplementary Fig. 3).

Collectively, these data show that compared with UPs, TPs and ECs do not express abnormally high levels of serum IFNα, avoiding the pathogenic IFNα effects on immune cells[14]. Particularly notable is the marked improvement in early effective NK cells and the later developing effective antigen-specific HLA-1–restricted CD8⁺ CTL and HLA-E–restricted CD8⁺supp cells in TPs and ECs. However, these improvements do not completely restore immune status to that of HDs.

## Discussion

Advances in HIV therapy have derived from understanding the molecular events of virus replication and targeting one or more stages with specific

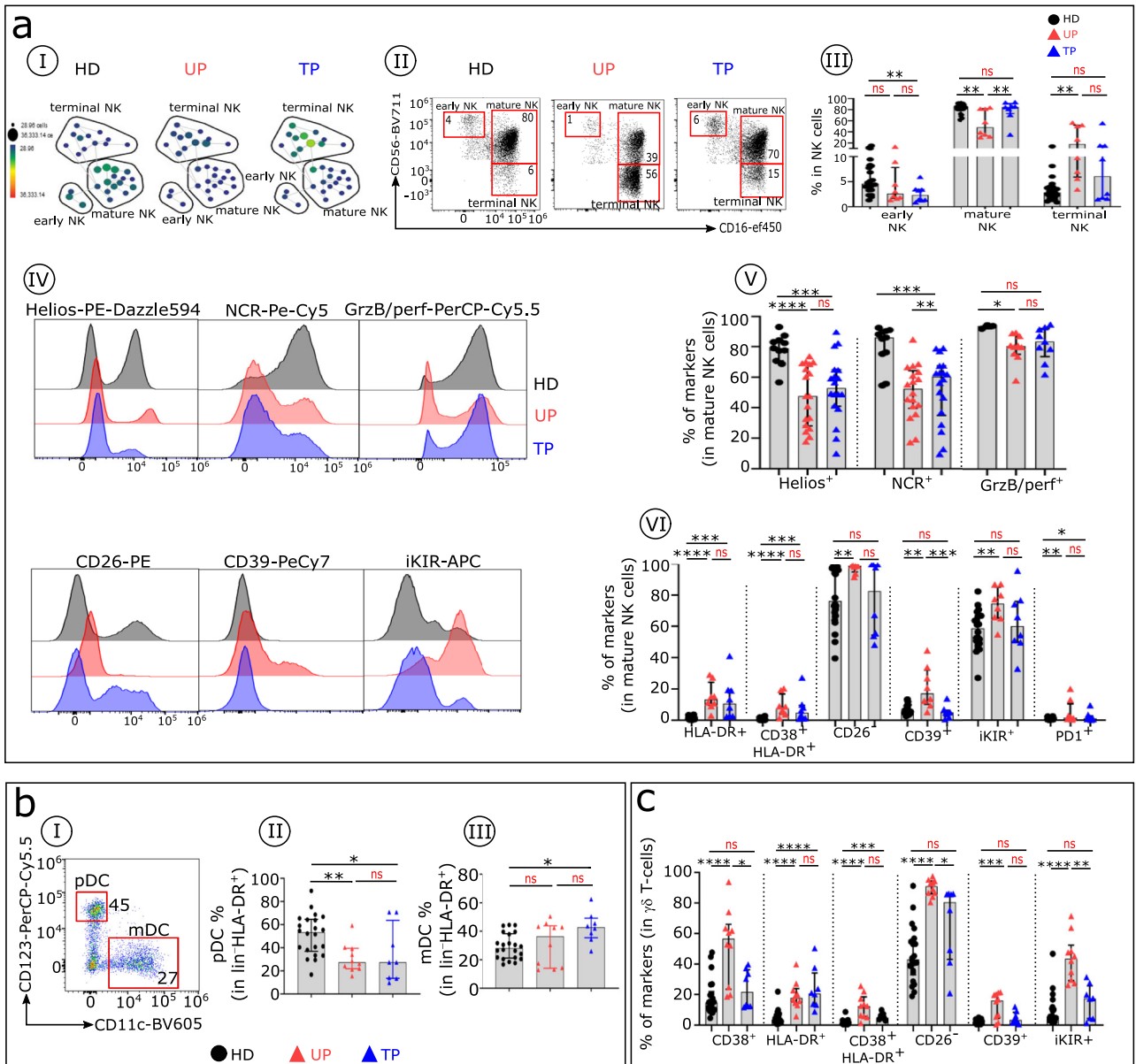

**Fig. 3 | TP distribution analysis of blood innate immune cells.** TPs had a lower percentage and fewer phenotypic alterations compared with UPs. **aI** The SPADE tree shows the distribution of NK cell subsets in HDs (black), UPs (red), and TPs (blue). Nodes are coloured by count. **aII** Representative flow cytometry plots of NK cell subsets gated on CD19⁻CD14⁻TCRγδ⁻CD3⁻HLA-DR⁻ cells from the three studied groups: early NKs (CD56^bright^CD16⁻), mature NKs (CD56^dim^CD16⁺), and terminal NKs (CD56⁻CD16⁺). **aIII** Frequency of early, mature, and terminal NKs in each studied group (HDs *n* = 22, UPs *n* = 10 and TPs *n* = 8). **aIV** Histograms with the expression level of Helios, NCR (NKp30, NKp44, NKp46), GrzB/perf, CD26, CD39, and iKIR on mature NK cells from HDs, UPs, and TPs. **aV, aVI** Boxplots display the frequency of the indicated markers in mature NK cells in each studied group.

Analysis was done in 22 HDs, 10 UPs, and 8 TPs for all markers except for Helios and NCR (11 HDs, 18 UPs, 18 TPs) and GrzB/perf (3 HDs, 10 UPs, 10 TPs). **bI** Representative dot plot shows how to distinguish pDC (CD123⁺ CD11C⁻) and mDC (CD123⁻CD11C⁺) subsets within the HLA-DR⁺ lin⁻ population in HD. Histograms with the frequencies of pDC (**bII**) and mDC (**bIII**) across the groups (HDs *n* = 22, UPs *n* = 10, and TPs *n* = 8). **c** Proportion of specific markers on TCR γδ T cells of each studied group (HDs *n* = 22, UPs *n* = 10, and TPs *n* = 8). Multiple group comparisons were made using the Kruskal–Wallis test with Dunn's multiple comparison testing. Values are medians and *p* values (**p* < 0.05, ***p* < 0.01, ****p* < 0.001, *****p* < 0.0001); ns: not significant. Error bars on graphs represent interquartile ranges.

drugs. However, therapy is needed throughout life for those who are not ECs. In addition to developing long-lasting therapies and perhaps targeting the integrated HIV proviral DNA of HIV reservoir cells, alternative approaches are needed. In agreement with the consortium of collaborators working on the ECs (EC consortium)[13], we think a deeper understanding of HIV pathogenesis and how these pathogenic mechanisms do not arise in ECs may be key to advances that could allow for >99% of infected non-EC patients to convert to EC status without treatment and possibly be functionally cured.

Abundant evidence shows that the key events leading to either HIV progression and AIDS or a favourable clinical outcome are predicated on the events of early infection[24,25]. What that early damage is and its prognostic underpinnings have been unclear. The first signs are the peak HIV viremia and subsequent virus set point. These develop during the initial stage of the immune response, at the time innate immunity kicks in. If not kept under tight control, a high HIV level leaves its mark on future developments. HIV kills infected CD4⁺T cells, chiefly after antigen activation[26]. A small percentage of macrophages also are

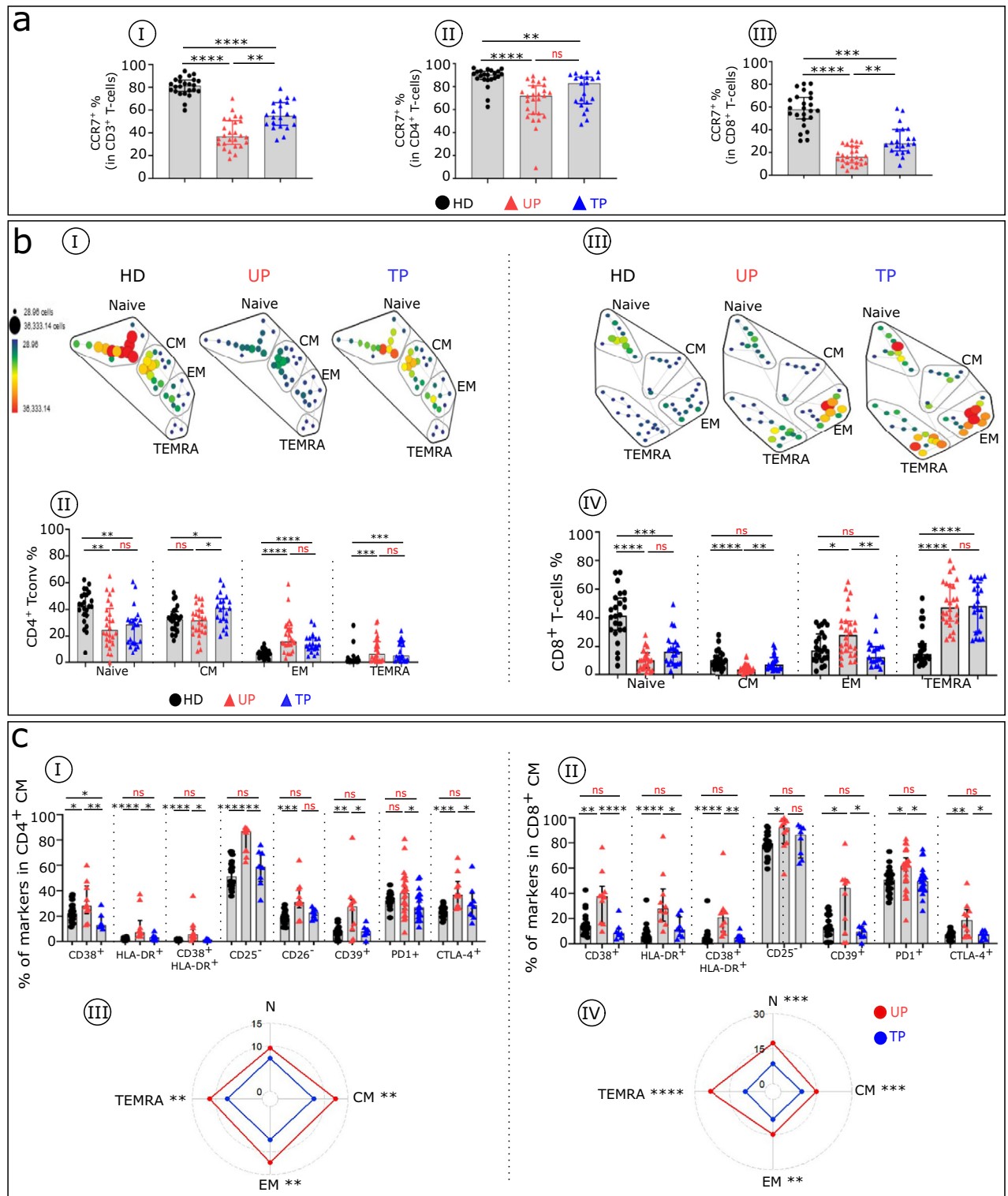

**Fig. 4 | Residual T cell phenotypic profile abnormalities in TPs. a** Histograms with the frequencies of CCR7 in CD3$^+$ (**aI**), CD4$^+$ (**aII**), and CD8$^+$ (**aIII**) T cells across the groups (HDs $n = 2$ (black), UPs $n = 26$ (red), and TPs $n = 22$ (blue)). **b** The SPADE tree shows the distribution of CD4$^+$T conv (**bI**) and CD8$^+$T cell (**bIII**) subsets in HDs, UPs, and TPs. Nodes are coloured by count. CD4$^+$Tconv and CD8$^+$T cells can be classified into four major subsets by their expression of CD45RA and the chemokine receptor CCR7: naïve (CCR7$^+$CD45RA$^+$), CM (CCR7$^+$CD45RA$^-$), EM (CCR7$^-$CD45RA$^-$), and TEMRA (CCR7$^-$CD45RA$^+$). The frequencies of naïve, CM, EM, and TEMRA CD4$^+$Tconv (**bII**) and CD8$^+$T cells (**bIV**) in each studied group (HDs $n = 22$, UPs $n = 26$, and TPs $n = 22$) are shown. Boxplots show the expression of the indicated marker in CD4$^+$CM (**cI**) and CD8$^+$CM (**cII**) cells across the groups (HDs $n = 22$, UPs $n = 10$, and TPs $n = 8$). The radar chart shows a composite score of phenotypic cell alteration calculated for each CD4$^+$Tconv (**cIII**) and CD8$^+$T cell (**cIV**) subpopulation in UPs (red lines) and TPs (blue lines) (see "Methods"). Multiple group comparisons were made using the Kruskal–Wallis test with Dunn's multiple comparison testing. Values are medians and *p* values (**$p < 0.05$, ***$p < 0.01$, ****$p < 0.001$, *****$p < 0.0001$); ns: not significant. Error bars on graphs represent interquartile ranges.

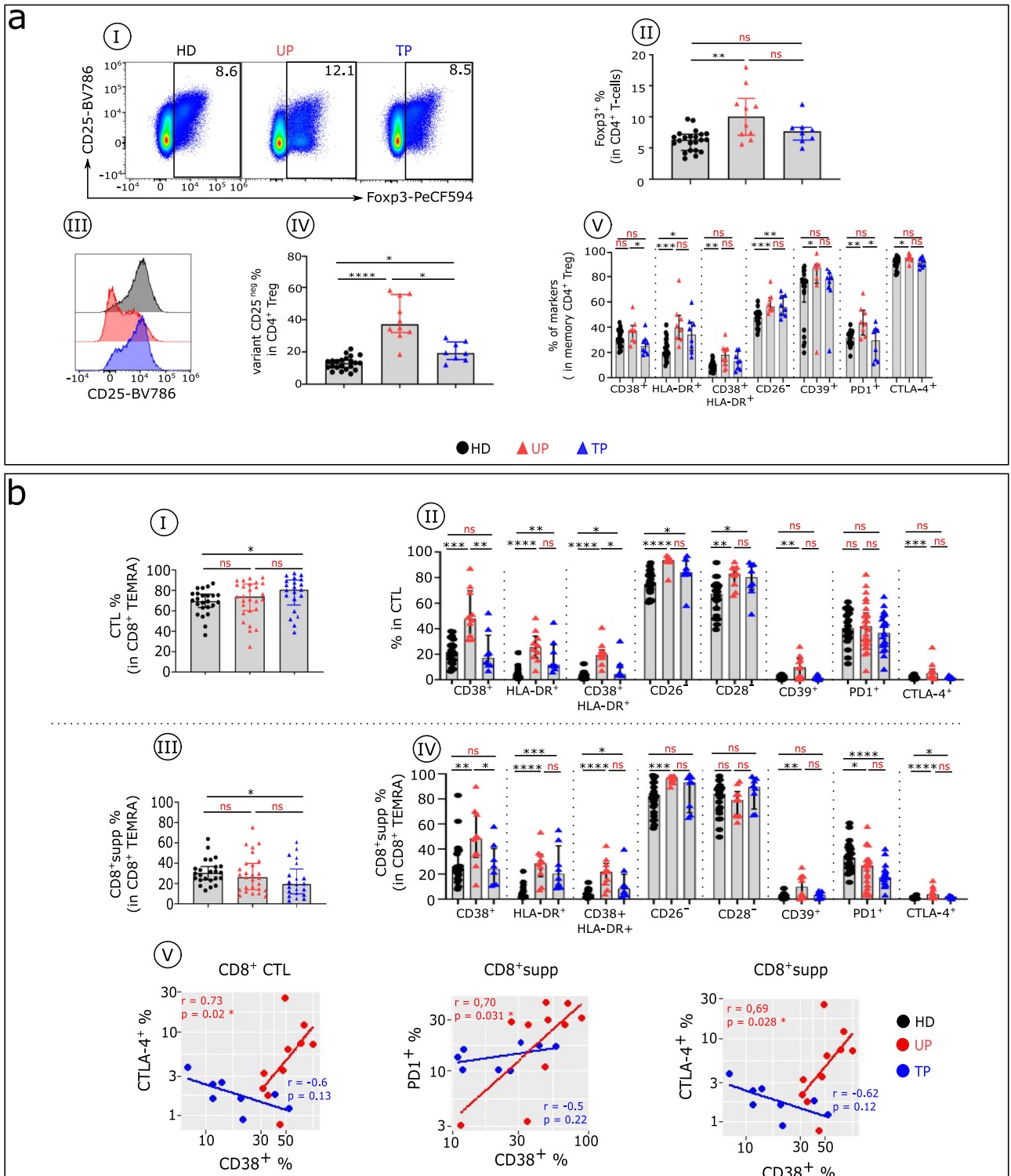

**Fig. 5 | Treg and cytotoxic CD8+T cells from TPs showing residual phenotypic abnormalities. aI** Representative flow cytometry plots of CD25+ Foxp3+ cells within CD4+T cells isolated from HDs (black), UPs (red), and TPs (blue). **aII** Histograms show the frequency of Foxp3 in CD4+T cells. **aIII** Histograms display the expression level of CD25 in CD4+ Foxp3+ T cells and (**aIV**) the frequency of the Treg CD25− variant in CD4+Foxp3 T cells in each studied group. **aV** Proportion of specific functional signalling checkpoints on memory CD4+Treg (CD4+ Foxp3+CD25+ CD45RA−) cells of each studied group (HDs $n = 22$, UPs $n = 10$, and TPs $n = 8$). **b** Histograms show the frequency of CD8+ CTL (CD8+TEMRA iKIR− (**bI**) and CD8+supp (CD8+TEMRA iKIR+) cells (**bIII**) in each studied group (HDs $n = 22$,

UPs $n = 26$, and TPs $n = 22$)). Boxplots show the proportion of specific markers on CD8+ CTL (**bII**) and CD8+supp (**bIV**) T cell subsets of each studied group (HDs $n = 22$, UPs $n = 10$, and TPs $n = 8$). **bV** Scatterplots of the relationships between the expression level of indicated markers in CD8+ cytotoxic T cells (CD8+ CTL and CD8+supp) (UPs $n = 10$ and TPs $n = 8$). Multiple group comparisons were made using the Kruskal–Wallis test with Dunn's multiple comparison testing, and correlations were analysed with Spearman's rank correlation test. Values are medians and $p$ values ($*p < 0.05$, $**p < 0.01$, $***p < 0.001$, $****p < 0.0001$); ns: not significant. Error bars on graphs represent interquartile ranges.

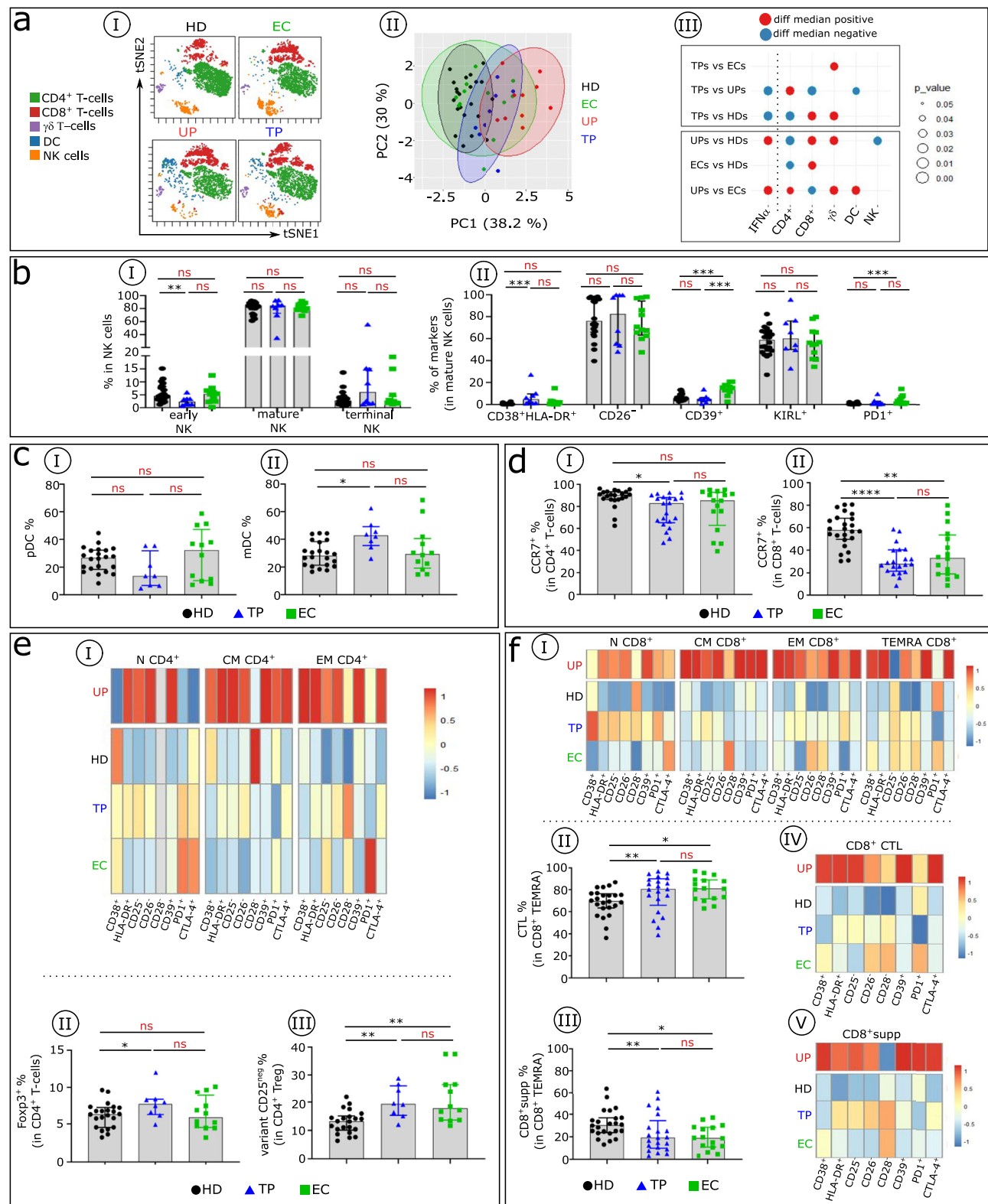

infected, but most avoid cytopathic effects[27], and other immune cells are not infected. It is evident, though, that there is much more to the story, and many reports have shown that despite being uninfected, other immune cells may not function normally. The implication is the existence of some unidentified critical mediator(s) of these effects, acting very early in infection, which excludes candidates involved in specific adaptive immune responses and the inflammatory cytokines associated

with chronic stages. Instead, this mediator must be active at the innate stage of the immune response (Supplementary Fig. 4).

Our results and previous reports lead us to conclude that the main mediator is elevated levels of IFNα[14]. Negative effects associated with IFNα have been described, but usually with the protein as one of several cytokines and acting in a general way to promote inflammation, as occurs in later stages of HIV infection. The details in our reports show that IFNα is more

**Fig. 6 | TPs and ECs share few blood immune cell anomalies. aI** Representative viSNE plot shows major immune cell subpopulation distribution (CD4[+], CD8[+], and TCR γδ T cells; NKs and DCs) in HDs (black), UPs (red), TPs (blue), and ECs (green), evaluated by flow cytometry. **aII** PCA scatterplots of samples based on the proportion of the different major lymphocyte subpopulations indicated above. Each point represents one participant, colour-coded by group: HDs (black), UPs (red), TPs (blue), and ECs (green). Each group is outlined by an ellipse representing the 95% confidence interval of the sample groupings. **aIII** The balloon plot summarizes the statistically significant changes in the indicated immune cell populations between each compared group. The size of the circle represents the *p* value. Red and blue indicate increased or decreased frequencies of immune cell populations. **bI** Frequency of early, mature, and terminal NKs in each studied group (HDs *n* = 22, TPs *n* = 8, and ECs *n* = 12). **bII** Boxplots display indicated marker frequency in mature NK cells in each studied group. Histograms show the frequencies of pDCs (**cI**) and mDCs (**cII**) across the groups (HDs *n* = 22, TPs *n* = 8, and ECs *n* = 12) and

CCR7 frequency in CD4[+] (**dI**) and CD8[+] (**dII**) T cells across the groups (HDs *n* = 22, TPs *n* = 8, and ECs *n* = 12). **eI** Heatmap of the indicated marker frequency in CD4[+]N, CM, and EM cells. **eII** Histograms show Foxp3 frequency in CD4[+]T cells and (**eIII**) Treg CD25[−] variant frequency in CD4[+]Foxp3 T cells in each studied group. **f I** Heatmap of the indicated marker frequency in CD8[+]N, CM, EM, and TEMRA cells. Histograms show the frequency of CD8[+] CTL (**fII**) and CD8[+]supp (**f III**) in each studied group (HDs *n* = 22, TPs *n* = 8, and ECs *n* = 12). Heatmap of the indicated marker frequency in CD8[+] CTL (**fIV**) and CD8[+]supp (**fV**) cells in each studied group (HDs *n* = 22, TPs *n* = 8, and ECs *n* = 12). Warmer colours indicate higher values and colder colours indicate lower values. Multiple group comparisons were made using the Kruskal–Wallis test with Dunn's multiple comparison testing. Values are medians and *p* values (\**p* < 0.05, \*\**p* < 0.01, \*\*\**p* < 0.001, \*\*\*\**p* < 0.0001); ns: not significant. Error bars on graphs represent interquartile ranges.

important in an early and specific disease stage. Other contributors to innate immunity are the NK cells. More than 99% of infected persons do not have adequate early control of HIV, so that HIV progression has a tailwind as time passes. As we describe in an accompanying paper[14] and in this report, both NK cells and IFNα show major abnormalities after infection in UPs. NK cells remain unaffected in ECs[14] and are markedly improved in non-EC patients after therapy, as shown here. In these non-EC, treated patients, NK cells exhibit specific features of normal NK cells, especially in the setting of HLA-B57. Undoubtedly, NK cells are key in the control of HIV in the earliest days after infection. IFNα levels are highly elevated from an early period in UPs, in direct correlation with the HIV titre[28,29]. In the specific case of HIV, IFNα is not an especially effective antiviral because it concomitantly induces high expression of the HIV co-receptor CCR5, as we show here (Fig. 2 and Supplementary Fig. 4). In addition, as we have previously reported, this treatment reduces HIV-inhibitory beta chemokines[20]. In the current work, we demonstrate that in untreated HIV-infected non-EC patients (i.e., UPs), all peripheral blood immune cells at all stages of the immune response show abnormalities in distribution numbers and immune phenotype. Of note, these abnormalities can be induced by elevated IFNα levels, as described in our companion paper and in other specific cases, as we and others have reported[30–32]. These special cases include induction of several inhibitory checkpoints and immunosuppressive molecules, diminishing of some essential positive immune signals, and interference with T cell homoeostasis and the subsequent adaptive immune response.

We show that ECs and also TPs to a lesser extent avoid these abnormalities of immune cell subsets (Fig. 6) and control IFNα levels and virus load[14] (Figs. 1 and 6aIII). The differences in immune cell anomalies between TPs and UPs are huge. Apart from the control of IFNα production in TPs and ECs, these differences particularly include[1] fewer infected CD4[+]T cells, likely because of the absence of IFNα-induced enhancement of CCR5 (Fig. 2)[2]; the presence of functional NK cells lacking any major IFNα-induced immune defects (Fig. 3a)[3]; low levels of IFNα-induced alterations in CD4[+]T helper and Treg cells (Figs. 4 and 5a); and[4] minimally altered antigen-specific cytotoxic CD8[+]T cells (both CTLs and CD8[+]supp T cells), which also lack the major IFNα-induced defects (Fig. 5b). We suggest that the remaining defects in cells from ECs and TPs reflect the residue of the early phase of the immune response, given the known thymic damage, whether IFNα-induced or not (Supplementary Fig. 4), and more so in TPs before cART[33,34].

IFNλ is a different story. Even in TPs, IFNλ remains elevated, suggesting the persistence of local mucosal tissue reservoirs of HIV. However, IFNλ does not have the detrimental effects on immune cells that IFNα does because immune CD4[+]T cells lack constitutive receptors for IFNλ[35], and this protein does not enhance CCR5 expression (Fig. 2a). Consequently, IFNλ may be a useful agent against HIV even when present at high levels.

Evidence indicates that genetic factors may confer partial protection in some ECs, particularly their HLA genetics and most especially HLA-B57[36]. Indeed, it is tempting to infer that these variants offer the explanation for the EC state because they are, of course, present at the onset of infection, and the

evidence of what dictates progression strongly favours an early event. However, these factors cannot explain the entirety of the EC group because many ECs do not carry the known genetic factors associated with EC and HLA-B57 is present in many non-EC HIV patients who experience typical disease progression[36]. A reduced CCR5 genetic trait also has been described but is present in only a small number of EC patients[37]. Our data argue against a single trait as determining EC status because their minimal abnormalities are numerous and diverse. Consequently, we have hypothesized the existence of an additional mechanism that also would be engaged at the onset: infection with a low inoculum of HIV. This factor would reduce the number of competing founder viruses and favour chances for a lower producer. From the onset, ECs would have low HIV input, which in turn would involve a low set point, low IFNα levels, and no negative impact on immune cell control of HIV. Although confirming this hypothesis presents a difficult and long-term challenge, the idea is testable in a few ways. The first relates to eclipse time (time from infection to detectable viremia) in ECs. In HIV-infected non-EC patients, this period is about 2 weeks, as shown by Robb and colleagues[38]. This hypothesis predicts a substantially longer eclipse time in ECs. Second, we may be able to use animal models to reproduce the proviral integration pattern in ECs with low-dose infection. Third, if the prediction is accurate, then patients infected with larger inoculums, such as haemophilia patients infected in the early years of HIV, would not have EC status.

HIV/AIDS is now a treatable disease, and for >99% of patients, therapy is a lifelong, sometimes with side-effects, co-morbidities, and possibly a shortened life span. A goal in the field is to develop a functional "cure" that precludes the need for lifelong therapy, but some HIV cell reservoirs with "silent" proviral DNA still may be re-awakened with therapy interruption or by external stimuli. Consequently, something more is needed. Based on our detailed knowledge of the molecular events in HIV replication, in our view, it is unlikely that many novel standard approaches remain to be explored other than longer lasting drugs and perhaps specific attacks on HIV proviral DNA. In agreement with the EC consortium[13] and its leadership, we suggest that a rewarding pursuit in the quest for a functional cure would be to focus on pathogenesis in the context of therapies that diminish HIV progression and imitate the EC state. In this regard, the work of the consortium defines a unique pattern of HIV integration, including spaced mono/oligoclonal clusters and HIV proviral DNA integration into silent DNA regions with a reduced frequency of escape mutations in cytotoxic epitopes and antibody contact regions[13,39]. As a result, they have suggested therapies geared toward reproducing this pattern in non-EC patients. We suggest that this pattern of HIV proviral integration may not be the original triggering event in control of HIV but rather the consequence of HIV control, which nonetheless subsequently helps maintain that control by markedly diminishing HIV expression. The key question then is, what is this original control? We hypothesize and our data indicate that it is prevention of elevated IFNα in the earliest stages of infection, whether through immediate therapy or by a low inoculum initiating infection.

IFNα is, of course, a key and immediate protector against foreign invaders. Sometimes, though, it contributes along with several other

cytokines to dangerous inflammation in the later stages of infection with some viruses. Our results here are directed to a newly characterized, early, and continuous solo pathogenic effect of IFNα. These results position increased IFNα as a key direct mediator of HIV pathogenesis, in addition to its known direct HIV effect. The IFNα effect develops along with HIV expansion early after infection and continues through progression to AIDS unless treatment is initiated, except in the case of ECs. As we hypothesize here, this IFNα effect can be avoided in such cases because of low-inoculum infection. We emphasize that beyond ART, therapy should include temporarily targeting elevated systemic IFNα as soon as HIV seropositivity is known. This strategy may offer a way to reduce measurable HIV proviral DNA intact sequences down to the EC level[14,39] while concomitantly providing non-harmful anti-viral IFNλ to control other viral infections. Our hope is that this approach will lead to the end of further need for ART, i.e., a functional cure, in keeping with the direction promoted by Walker and colleagues[13].

## Data availability

Source data for the main figures in the manuscript can be accessed at figshare https://doi.org/10.6084/m9.figshare.25125356[40].

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

## Acknowledgements

We thank the study volunteers and medical personnel. We also thank Elissa Miller for skilful administrative and editorial collaboration and Philip Embiricos for financial support sponsorship. M.M.S. was supported by R01AI147870 and IBX002358A.

## Author contributions

R.G., D.Z., and H.L.B. designed the research; H.L.B., V.S., M.M., D.B., A.H.-K., and H.S. performed the research; H.L.B., V.S., M.M., D.B., A.H.-K., J.-D.B., A.B., A. Burny, R.G., and D.Z. analysed the data; S.K., M.S., C.F.L., and G.D. provided blood cells and serum samples, clinical data samples from HIV patients, and clinical input; and H.L.B., R.G., and D.Z. wrote the paper. All authors contributed to reviewing and editing and agreed to the published version of the manuscript.

## Competing interests

The authors declare no competing interests.

## Additional information

[1]Université de Paris; INSERM U976, HIPI Unit, Institut de Recherche Saint-Louis, F-75010 Paris, France. [2]Laboratory for Genomics Foundation Jean Dausset-CEPH, Paris, France. [3]Institute of Human Virology, School of Medicine, University of Maryland, Baltimore, MD 21201, USA. [4]Laboratory of Infectious Diseases, GIGA-I3, GIGA-Institute University of Liege, 4000 Liege, Belgium. [5]Dermatology Department, Hôpital Saint-Louis, Assistance Publique-Hôpitaux de Paris (AP-HP), Paris, France. [6]Laboratory of Molecular Biology, Gembloux Agrobiotech, University of Liège, Liège, Belgium. [7]Global Virus Network, Baltimore, MD 21201, USA. [8]University of Maryland School of Medicine, Baltimore, MD 21201, USA. [9]Program in Oncology, Marlene and Stewart Greenebaum Comprehensive Cancer Center, University of Maryland, Baltimore, MD 21201, USA. [10]21CBIO, Paris, France. [11]These authors contributed equally: Hélène Le Buanec, Valérie Schiavon, Marine Merandet. [12]These authors jointly supervised this work: Daniel Zagury, Robert C. Gallo. ✉e-mail: rgallo@ihv.umaryland.edu

