## [Peer Review File · Communications Medicine]

Reviewers' comments:

Reviewer #1 (Remarks to the Author):

The manuscript from Buanec and colleagues attempts to causally link elevated interferon levels with HIV disease outcomes. The authors report higher levels of IFN alpha in untreated subjects as opposed to treated subjects and Elite Controllers. For the most part, the analysis is comprehensive and well controlled. Addressing the following concerns may help overall clarity.

1: The authors argue for a causative role of IFN alpha in disease progression. The increased levels of IFN alpha in untreated subjects isn't surprising and has been shown by others. The authors should clarify whether IFN alpha is being driven by viral replication or do they have another explanation for the relationship between IFN alpha levels and HIV in untreated infection.

2: Do the authors have data on the IFN alpha levels at peak or at set-point for the subjects in their cohort? If so, a correlation between IFN alpha at peak / set point and progression would support a causative role as opposed to collateral role for IFN in disease progression.

3: If, as the authors suggest, part of the mechanism is induction of CCR5 by IFN alpha and creation of more substrates for HIV replication, have they looked at the relative abundances of R5 versus X4 viruses in post- treatment interruption viremia?

Reviewer #2 (Remarks to the Author):

Le Buanec and colleagues performed a comprehensive immunophenotyping with up-to-date bioinformatics analysis in untreated (UP, n=26) and treated HIV-infected patients (TP, n=21), uninfected donors (HD, n=24), and elite controllers (EC, n=12), focusing in this and the companion manuscript on interferon (IFN- α)-induced alterations.

Mains findings: The authors show significantly elevated peripheral IFN- α , but not IFN- λ 2 levels using SIMOA technology in UP compared to HD and TP. CCR5 mRNA and protein expression was upregulated in stimulated CD4+ T-cells upon exposure to increasing IFN- α concentrations. Pretreatment of stimulated CD4+ T cells (HD) with IFN- α reduced the inhibitory effect on a CCR5-, but not on a CXCR4-tropic virus. The count of NK cell subsets and their phenotype was altered in UP and not completely reversed in TP compared to HD. Similar results were obtained with distributions of CD4 and CD8 T-cell subsets and their expression of markers for activation and differentiation. In UP, Treg counts were increased, as were CTL and CD8+supp cells, which expressed increased markers of activation, differentiation and exhaustion. Finally, the authors show similarities in counts and phenotype of nearly all immune cell types between EC and TP.

The authors conclude that uncontrolled HIV-1 infection increases IFN- α secretion, which promotes further viral propagation via increased expression of coreceptor CCR5. Limiting this IFN- α secretion may contribute to a functional "cure" in HIV-1 infection. Early control of HIV-1 infection by cells of the innate immune defense, in particular NK cells, could reduce IFN-alpha production, contributing to elite controller status.

I do embrace the concept that elevated IFN- α levels play an important role in the HIV-1 immunopathogenesis. However, this concept is not new and has already been published by others (e.g. Jay Levy's group), which should be cited (PMID 16278001, 22203858). Importantly, the authors show that phenotypic alterations that are present in UP and are partially reversed in TP, can be induced in vitro by IFN- α .

Major points:

- The amount of peripheral IFN- α levels in UP ranges between 10-1000 fg/ml (Fig. 1D). The lowest IFN- α concentration that induces CCR5 mRNA upregulation was shown to be 10 pg/ml (Fig. 1A), while blocking of p24 secretion started with 0.2 ng/ml (Fig. 2C). Why are much higher IFN- α concentrations required for morphological changes in vitro? This is even more puzzling as UP have lower CCR5 on the CD4+ T-cell surface (at 0 ng/ml IFN- α) compared to HD, EC and TP (Fig. 2B). Following the authors' reasoning, shouldn't UP have the highest CCR5 levels at baseline?
- Fig. 1A: is CXCR4 mRNA and protein expression also upregulated by IFN- α ?
- Fig. 2C: These data are key to the manuscript. No statistics was performed. The experiment should be repeated in another two to three donors and then the data of all donors should be summarized. Use different MOI to show the effect of IFN- α on the infection dose.
- Statistics: which post-hoc test was used to control for multiple testing? What is meant by "biologically independent samples"?
- How long were the TP actually treated at the time of sampling? It may make a difference for what period of time viral load was undetectable.

Minor points:

- Methods, human samples: please provide reference numbers for ethical approval.
- Results, p. 5: The authors say that DC are increased in TP and UP compared to HD. Please specify whether PDC or MDC are meant here (also in Fig. 1B5). The percentage of PDC and MDC was calculated with respect to lin- HLA-DR+ cells (and not PBMC), which should be clearly indicated throughout the manuscript.
- Discussion: The authors may want to speculate whether elevated IFN-alpha levels in UP cause the switch of CCR5- towards CXCR4-tropic viruses in the course of disease?
- Fig. 6: The authors show data of 12 EC, but the Extended data in Table 3 list 18 EC.
- Abbreviations need to be better explained in the figure and table legends. Extended data Table 1: (med) means (median)? Extended data Table 3a: HIV dx? AA? Year of sample date – 012, 011? Table 3b: Excel sheet should be converted into a regular table. Figs. 6E1, 6F1, 6F4, 6F5: it is not clear what is meant by the scale. Is the cohort of UP or the highest value in all cohorts set as "1"?
- Both manuscripts access the same patients and partially overlap. On some occasions, data referred to in this manuscript (e.g. IFN- α levels in EC) are shown in the companion manuscript only. These

parts need to be properly referenced.

- The idea that elite controllers were infected by chance with a low viral load that does not induce 'vicious IFN- α production' is interesting but speculative. Conversely, EC might efficiently control HIV-1 infection in the short and (!) long run and therefore not respond with increased IFN-alpha production. To this end, it might be helpful to examine samples from seroconverters, which should be discussed.

Answer to Paper 2 reviewers' comments

Reviewer #1 (Remarks to the Author):

The manuscript from Buanec and colleagues attempts to causally link elevated interferon levels with HIV disease outcomes. The authors report higher levels of IFN alpha in untreated subjects as opposed to treated subjects and Elite Controllers. For the most part, the analysis is comprehensive and well controlled. Addressing the following concerns may help overall clarity.

1: The authors argue for a causative role of IFN alpha in disease progression. The increased levels of IFN alpha in untreated subjects isn't surprising and has been shown by others. The authors should clarify whether IFN alpha is being driven by viral replication or do they have another explanation for the relationship between IFN alpha levels and HIV in untreated infection.

Response : We did not have the opportunity to follow serum IFN α level and the HIV viral load in HIV-infected patients from the onset of their infection. Other teams had this opportunity. Hardy GA *et al* reported that, in untreated HIV-infected patients, plasma IFN α levels correlated positively with plasma HIV-1 RNA levels (Hardy GA et al, Interferon- α is the primary plasma type-I IFN in HIV-1 infection and correlates with immune activation and disease markers. PLoS One. 2013;8(2):e56527. doi: 10.1371/journal.pone.0056527. Epub 2013 Feb 20. PMID: 23437155; PMCID: PMC3577907). Indeed, elevated serum IFN α concentration is a primary serological abnormality reported in AIDS ((DeStefano E et al. Acid-labile human leukocyte interferon in homosexual men with Kaposi's sarcoma and lymphadenopathy. J Infect Dis. 1982 Oct;146(4):451-9. doi: 10.1093/infdis/146.4.451. PMID: 7119475). Stacey AR et al also analyze kinetics changes in serum IFN α level and viral load in sequential samples collected from plasma donors acquiring HIV infections during the earliest stages of this infection. They show that that IFN α is detected only after viremia, and that IFN α level is associated with viral load (Stacey AR et al. Induction of a striking systemic cytokine cascade prior to peak viremia in acute human immunodeficiency virus type 1 infection, in contrast to more modest and delayed responses in acute hepatitis B and C virus infections. J Virol. 2009 Apr;83(8):3719-33. doi: 10.1128/JVI.01844-08. Epub 2009 Jan 28. PMID: 19176632; PMCID: PMC2663284.)

2: Do the authors have data on the IFN alpha levels at peak or at set-point for the subjects in their cohort? If so, a correlation between IFN alpha at peak / set point and progression would support a causative role as opposed to collateral role for IFN in disease progression.

Response : We did not have the opportunity to follow IFN α level in HIV-infected patients from the onset of their infection.

However, comparisons of SIV infection in primate species that develop AIDS-like disease and species without disease symptoms indicate that an elevated IFN α serum concentration overflowing from lymphoid tissues to body fluid occur only during pathogenic infection in macaques, whereas natural SIV hosts, without disease progression, express residual background IFN α concentration levels (Mandl, J. N. et al. Divergent TLR7 and TLR9 signaling and type I interferon production distinguish pathogenic and nonpathogenic AIDS virus infections. *Nature Med.* 14, 1077–1087 (2008) ; Jacquelin, B. et al. Nonpathogenic SIV infection of African green monkeys induces a strong but rapidly controlled type I IFN response. *J. Clin. Invest.* 119, 3544–3555 (2009)). Similar findings have been made in individuals infected with HIV; rapid progressors show higher IFN α signatures than viremic non-progressors (Rotger, M. et al. Comparative transcriptomics of extreme phenotypes of human HIV-1 infection and SIV infection in sooty mangabey and rhesus macaque. *J. Clin. Invest.* 121, 2391–2400 (2011)).

It is noteworthy, as indicated in the revised companion paper, to point out that IFN α , like other cytokines produced in tissues of healthy individuals have a short half-life and express their message locally in a paracrine/juxtacrine manner on target cells expressing type I IFN α receptor. However, when abnormally produced, elevated level of IFN α , which overflow diluted into body fluid with concentration ranging from 10-1000 fg in UP serum.

Our data and these studies suggest that the initial dose of virus at infection drives the elevated level of IFN α and its pathogenesis: if the infection occurs in the presence of an elevated dose of virus, subsequently there will be a correlated elevated dose of IFN α , and a progression of IFN α pathogenesis leading to AIDS.

3: If, as the authors suggest, part of the mechanism is induction of CCR5 by IFN alpha and creation of more substrates for HIV replication, have they looked at the relative abundances of R5 versus X4 viruses in post- treatment interruption viremia?

Response : It is difficult to answer, considering that the rebound virus can originate from several cellular and anatomical compartments after treatment interruption (De Scheerder MA et al. HIV Rebound Is Predominantly Fueled by Genetically Identical Viral Expansions from Diverse Reservoirs. Cell Host Microbe. 2019 Sep 11;26(3):347-358.e7. doi: 10.1016/j.chom.2019.08.003. Epub 2019 Aug 27. PMID: 31471273)

On our observations so far, the R5 viruses are almost always predominant in circulating plasma. This could be mainly because R5 and X4 viruses are replicating in different CD4 subsets. While R5 viruses are mainly in effector memory and terminal TEMRA subsets compartments, whereas the X4 viruses preferentially target the central memory and naive subsets in lymphoid tissues. The different viral burst size (higher for the more activated/differentiated CD4⁺ T-cells where the R5 viruses replicate) provide R5 viruses advantage in body fluid (plasma).

In the revised paper, we added in Fig. 2 our experimental data showing that CCR5 but not CXCR4 expression in CD4⁺ T-cell cultured in presence of IFN α is enhanced in a dose effect manner (Fig. 2A3 and 2A4. We have also added the following sentence to the results section, at the end of the paragraph (penultimate line) entitled "High IFN α level enhances *in vitro* expression of the HIV coreceptor CCR5":

Interestingly, IFN α has no effect on CXCR4 expression at either mRNA level (Fig. 2A3) and protein level (Fig. 2A4) on stimulated CD4⁺ T-cells.

Reviewer #2 (Remarks to the Author):

Le Buanec and colleagues performed a comprehensive immunophenotyping with up-to-date bioinformatics analysis in untreated (UP, n=26) and treated HIV-infected patients (TP, n=21), uninfected donors (HD, n=24), and elite controllers (EC, n=12), focusing in this and the companion manuscript on interferon (IFN)- α -induced alterations.

Mains findings: The authors show significantly elevated peripheral IFN- α , but not IFN- λ 2 levels using SIMOA technology in UP compared to HD and TP. CCR5 mRNA and protein expression was upregulated in stimulated CD4⁺ T-cells upon exposure to increasing IFN- α concentrations. Pretreatment of stimulated CD4⁺ T cells (HD) with IFN- α reduced the inhibitory effect on a CCR5-, but not on a CXCR4-tropic virus. The count of NK cell subsets and their phenotype was altered in UP and not completely reversed in TP compared to HD. Similar results were obtained with distributions of CD4 and CD8 T-cell subsets and their expression of markers for activation and differentiation. In UP, Treg counts were increased, as were CTL and CD8+supp cells, which expressed increased markers of activation, differentiation and exhaustion. Finally, the authors show similarities in counts and phenotype of nearly all immune cell types between EC and TP.

The authors conclude that uncontrolled HIV-1 infection increases IFN- α secretion, which promotes further viral propagation via increased expression of coreceptor CCR5. Limiting this IFN- α secretion may contribute to a functional “cure” in HIV-1 infection. Early control of HIV-1 infection by cells of the innate immune defense, in particular NK cells, could reduce IFN-alpha production, contributing to elite controller status.

I do embrace the concept that elevated IFN- α levels play an important role in the HIV-1 immunopathogenesis. However, this concept is not new and has already been published by others (e.g. Jay Levy’s group), which should be cited (PMID 16278001, 22203858). Importantly, the authors show that phenotypic alterations that are present in UP and are partially reversed in TP, can be induced in vitro by IFN- α .

Response : We already answered this question in our response to the companion paper (please see this response).

The two manuscripts have not the goal to report that serum IFN α concentration is pathogenically elevated (observation already reported initially by us and confirmed by multiple reports as in those from Shearer’s and Levy ‘s groups. The present manuscript aimed at showing that besides HIV itself, elevated IFN α is a key mediator of HIV pathogenesis.

Nevertheless, to satisfy the referee's request, we have added, in addition to the Shearer's group papers, the paper by Levy's group in the revised accompanying document, both of which confirm our 1990' ones showing the pathogenicity of elevated IFN α in HIV infection.

Major points:

- The amount of peripheral IFN- α levels in UP ranges between 10-1000 fg/ml (Fig. 1D). The lowest IFN- α concentration that induces CCR5 mRNA upregulation was shown to be 10 pg/ml (Fig. 1A), while blocking of p24 secretion started with 0.2 ng/ml (Fig. 2C). Why are much higher IFN- α concentrations required for morphological changes in vitro? This is even more puzzling as UP have lower CCR5 on the CD4+ T-cell surface (at 0 ng/ml IFN- α) compared to HD, EC and TP (Fig. 2B). Following the authors' reasoning, shouldn't UP have the highest CCR5 levels at baseline?

Response : As other cytokines, IFN α has short half-life, is produced in tissues and particularly in lymphoid organs. IFN α acts on their target cells locally and is released into the body fluid at only background level (< 100 fg/ml in HD but also in most of EC and TP). However, when abnormally highly produced in lymph nodes, as in UP, IFN α abnormally overflows into the body fluid and becomes diluted in serum (at concentration of 10-1000 fg/mL). This serum concentration is obviously much lower than that present in tissues where they exert their action through specific high-affinity receptors (with Kd \approx 10–10 nM). The detection of the abnormal presence of IFN α in the serum can only be achieved with an ultrasensitive digital ELISA (Simoa).

During acute HIV infection, IFN α produced in lymph nodes, contributes to antiviral protection by induction of an ISG-based cellular antiviral program. The level of IFN α produced is correlated with the viral load. If the infection occurs in the presence of a high amounts of virus, subsequently there will be a production of an elevated level of IFN α , that is abnormally overflowing into body fluids is pathogenic.

The concentrations of IFN α used in CD4⁺ T-cells and NK cells culture correspond to that present in tissues, and not to that abnormally found in serum of UP. We would agree with the comment of referee 2 who considers it puzzling that UP have lower CCR5 in at the CD4⁺ T-cell surface, if we did not consider untreated patients with high viral loads whose viral particles bind to their CCR5 receptor resulting in internalization of this receptor.

- *Fig. 1A: is CXCR4 mRNA and protein expression also upregulated by IFN- α ?*

Response : When we measure the effect of IFN α on CXCR4 expression at protein and mRNA level, we do not observe an increase in CCR4 expression, but rather a slight decrease that varies according to the donor.

In the revised manuscript, we have added to the Fig. 2, new IFN α data on the expression of the two chemokine receptors CCR5 and CXCR4, confirming this claim (Fig. 2A3 and 2A4). We have also added the following sentence to the results section, at the end of the paragraph (penultimate line) entitled "High IFN α level enhances in vitro expression of the HIV coreceptor CCR5": Interestingly, IFN α has no effect on CXCR4 expression at either mRNA level (Fig. 2A3) and protein level (Fig. 2A4) on stimulated CD4⁺ T-cells.

- Fig. 2C: These data are key to the manuscript. No statistics was performed. The experiment should be repeated in another two to three donors and then the data of all donors should be summarized. Use different MOI to show the effect of IFN- α on the infection dose.

Response : Result shown in Fig.2C is typical of the 3-patient study. This is an illustrative experiment of the summary. Using different MOI to show the effect of IFN α on the dose of infection is interesting, but If MOI are high, of course, IFN α would not show visible effects, and of course too low and the experiment becomes impossible. We selected a doable MOI where infection can be seen, IFN α inhibitory effects may be noted (in CXCR4 HIV), and the decline in inhibition is evident in CCR5 HIV.

- *Statistics: which post-hoc test was used to control for multiple testing? What is meant by “biologically independent samples”?*

Response : Comparisons between groups were performed with unpaired nonparametric Mann-Whitney as indicated in the statistical analyses section of the material and methods of the manuscript.

In the revised manuscript, comparisons between two groups are performed with unpaired nonparametric Mann-Whitney and comparisons among multiple groups are performed using Kruskal–Wallis test followed by Dunn’s Post-hoc test. Methods of analysis were also indicated in the statistical analyses section of the material and methods in the revised text and in the figure legends.

Please accept our apologies for the lack of clarity for the sentence “biologically independent samples”? We simply wanted to say that each individual sample was analyzed once for each of the parameters studied.

So we remove “**biologically independent samples**” in the revised text.

- *How long were the TP actually treated at the time of sampling? It may make a difference for what period of time viral load was undetectable.*

Response : All TP received ART for at least 3 years and had HIV VL suppressed throughout this time.

Minor points:

- Methods, human samples: please provide reference numbers for ethical approval.

Response : The IRB number for the study in US is H-29331. In Liege (Belgium) the number of the agreement for the study is 2020/418. We added it in the material and method in the human samples section.

- Results, p. 5: The authors say that DC are increased in TP and UP compared to HD. Please specify whether PDC or MDC are meant here (also in Fig. 1B5). The percentage of PDC and MDC was calculated with respect to lin- HLA-DR+ cells (and not PBMC), which should be clearly indicated throughout the manuscript.

Response : In the Fig. 1B5, we present the frequency of DC in blood immune cells. However, indeed, the percentage of pDC and mDC was calculated with respect to lin- HLA-DR+ cells. We have now specified it in fig. 3B and in the legend as follows : **“Representative dot plot showing how to distinguish pDC (CD123+ CD11C-) and mDC (CD123-CD11C+) subsets within the HLA-DR+ lin- population in HD. Histograms showing the frequencies of pDC (B2) and mDC (B3) across the groups (HD n=22, UP n=10 and TP n=8)”**

- Discussion: The authors may want to speculate whether elevated IFN-alpha levels in UP cause the switch of CCR5- towards CXCR4-tropic viruses in the course of disease?

Response : We have no reason to speculate that high IFN α would change co-receptor although it is an interesting concept.

- Fig. 6: The authors show data of 12 EC, but the Extended data in Table 3 list 18 EC.

Response : To perform our phenotypic analysis, we had to prepare 2 cytometry panels. For some EC

patients, we lacked cells to perform the two panels, which is why in some figures there are 18 EC and in others 12.

- Abbreviations need to be better explained in the figure and table legends. Extended data Table 1: (med) means (median)? Extended data Table 3a: HIV dx? AA? Year of sample date – 012, 011? Table 3b: Excel sheet should be converted into a regular table. Figs. 6E1, 6F1, 6F4, 6F5: it is not clear what is meant by the scale. Is the cohort of UP or the highest value in all cohorts set as “1”?

Response : We removed the abbreviations in the titles of the legends and indicated in the legend what the abbreviation corresponds to when it appears the first time.

We modify the text as follows :

Legend Fig.1 : Title : Comparative analysis of major blood immune cell subsets and serum IFNs concentrations in Untreated Patients, Treated Patients and Healthy Donors.

Legend Fig.3 : NCR, GrzB/perf and iKIR were replaced by Natural cytotoxicity receptors, Granzyme/Perforin and Inhibitory Killer Ig-Like Receptors.

Concerning the Year of sample date, (– 012, 011), thank you for pointing out the typo. We fixed it in the revised manuscript text the date as follows: 2012, 2011

In extended table 1, we have added a legend to explain the abbreviations: NA : Not Available; med : median

In the extended tables describing the patients we have added a legend to explain the abbreviations: Dx : Diagnosis, M : Male, F : Female, AA : Afro American, IDU : Injection Drug Use, HS : Homosexual, MSM : Men who have Sex with Men, NA : Non Available.

As to the scale in Figs. 6E1, 6F1, 6F4, 6F5, the highest value in all cohorts is set as “1”.

- Both manuscripts access the same patients and partially overlap. On some occasions, data referred to in this manuscript (e.g. IFN- α levels in EC) are shown in the companion manuscript only. These parts need to be properly referenced.

Response : In order to facilitate the reading of the text, we have added in the 1st line of the 4th paragraph of the discussion chapter, the reference of the companion paper, reference 14, when we speak of the serum level of IFN α in ECs in the revised manuscript as follows :

“We show that EC and, to a lesser extent TP avoid these abnormalities of immune cell subsets (Figs. 2-6), and control IFN α levels and virus load (14) (Fig. 1 and Fig. 6A3)”

- The idea that elite controllers were infected by chance with a low viral load that does not induce ‘vicious IFN- α production’ is interesting but speculative. Conversely, EC might efficiently control HIV-1 infection in the short and (!) long run and therefore not respond with increased IFN-alpha production. To this end, it might be helpful to examine samples from seroconverters, which should be discussed.

Response : We of course agree with the reviewer on this point also questioned in the first manuscript. However, please keep in mind this is a hypothesis not meant to be solved here. The contribution of low infectious inoculum to the control of elevated IFN α is one hypothesis based on the following items:

- 1) Absence of the unique genetic parameter including HLA-B57, CCR5 Δ 32 and others;
- 2) Heterogeneous cytotoxic capacity of NK, CTL and CD8⁺ supp T-cells, variable between each EC observed in this study (text and Fig. 5 of the companion paper). We were struck by the fact that EC experts did not bring this up (the size of HIV inoculum) as far we know ever before because for us it seems the very likely answer that explains EC status.

3) Sequential analysis of sera from untreated HIV patients from the onset of infection which shows that a) IFN α is detected in serum after viral load detection, and b) serum IFN α levels correlate with viral load

levels (Hardy GA et al. Interferon- α is the primary plasma type-I IFN in HIV-1 infection and correlates with immune activation and disease markers. *PLoS One*. 2013;8(2):e56527. doi: 10.1371/journal.pone.0056527. Epub 2013 Feb 20. PMID: 23437155; PMCID: PMC3577907; Stacey AR et al. Induction of a striking systemic cytokine cascade prior to peak viremia in acute human immunodeficiency virus type 1 infection, in contrast to more modest and delayed responses in acute hepatitis B and C virus infections. *J Virol*. 2009 Apr;83(8):3719-33. doi: 10.1128/JVI.01844-08. Epub 2009 Jan 28. PMID: 19176632; PMCID: PMC2663284.)

4) It is well known that patients with high viral load “set points” experience disease progression much more rapidly than those with lower viral load set points. Elite controllers, therefore, represent the extreme of those with low viral load set points, corresponding to correlated low IFN α production, sufficiently controlled during the acute infection by innate immune defense mechanisms (IFN α , IFN λ and NK-cells) and during adaptive phase by additional cytotoxic cells. (Goujard C et al. Spontaneous control of viral replication during primary HIV infection: when is “HIV controller” status established? *Clin Infect Dis*. 2009 Sep 15;49(6):982-6. doi: 10.1086/605504. PMID: 19681706; Okulicz JF et al. Infectious Disease Clinical Research Program (IDCRP) HIV Working Group. Clinical outcomes of elite controllers, viremic controllers, and long-term nonprogressors in the US Department of Defense HIV natural history study. *J Infect Dis*. 2009 Dec 1;200(11):1714-23. doi: 10.1086/646609. PMID: 19852669; Woldemeskel BA et al. *EBioMedicine*. 2020 Dec;62:103118. PMID: 33181459; PMCID: PMC7658501)

5) Some Elite controller lost their Elite controller status especially due to an episode of viral superinfection associated with high IFN α production. We have seen that two elite controllers in our cohort lost their status (EC11 and EC52) following HCV infection associated with an increase in antiviral IFN α . A study published by Rossol S et al also shows, that a high presence of high levels of serum IFN α , notably, preceded the loss of virological control in elite controller (Rossol S et al Interferon production in patients infected with HIV-1. *J Infect Dis*. 1989 May;159(5):815-21. doi: 10.1093/infdis/159.5.815. PMID: 2468718).

All of this is now discussed in the revised manuscript.

REVIEWERS' COMMENTS:

Reviewer #1 (Remarks to the Author):

The authors have adequately addressed my concerns

Reviewer #2 (Remarks to the Author):

I appreciate the responses of the authors and, in particular, the additional experiment investigating the effect of IFN-alpha on CXCR4 expression.

Answer to Paper 2 reviewers' comments

Reviewer #1 (Remarks to the Author):

The manuscript from Buanec and colleagues attempts to causally link elevated interferon levels with HIV disease outcomes. The authors report higher levels of IFN alpha in untreated subjects as opposed to treated subjects and Elite Controllers. For the most part, the analysis is comprehensive and well controlled. Addressing the following concerns may help overall clarity.

1: The authors argue for a causative role of IFN alpha in disease progression. The increased levels of IFN alpha in untreated subjects isn't surprising and has been shown by others. The authors should clarify whether IFN alpha is being driven by viral replication or do they have another explanation for the relationship between IFN alpha levels and HIV in untreated infection.

Response : We did not have the opportunity to follow serum IFN α level and the HIV viral load in HIV-infected patients from the onset of their infection. Other teams had this opportunity. Hardy GA *et al* reported that, in untreated HIV-infected patients, plasma IFN α levels correlated positively with plasma HIV-1 RNA levels (Hardy GA et al, Interferon- α is the primary plasma type-I IFN in HIV-1 infection and correlates with immune activation and disease markers. PLoS One. 2013;8(2):e56527. doi: 10.1371/journal.pone.0056527. Epub 2013 Feb 20. PMID: 23437155; PMCID: PMC3577907). Indeed, elevated serum IFN α concentration is a primary serological abnormality reported in AIDS ((DeStefano E et al. Acid-labile human leukocyte interferon in homosexual men with Kaposi's sarcoma and lymphadenopathy. J Infect Dis. 1982 Oct;146(4):451-9. doi: 10.1093/infdis/146.4.451. PMID: 7119475). Stacey AR et al also analyze kinetics changes in serum IFN α level and viral load in sequential samples collected from plasma donors acquiring HIV infections during the earliest stages of this infection. They show that that IFN α is detected only after viremia, and that IFN α level is associated with viral load (Stacey AR et al. Induction of a striking systemic cytokine cascade prior to peak viremia in acute human immunodeficiency virus type 1 infection, in contrast to more modest and delayed responses in acute hepatitis B and C virus infections. J Virol. 2009 Apr;83(8):3719-33. doi: 10.1128/JVI.01844-08. Epub 2009 Jan 28. PMID: 19176632; PMCID: PMC2663284.)

2: Do the authors have data on the IFN alpha levels at peak or at set-point for the subjects in their cohort? If so, a correlation between IFN alpha at peak / set point and progression would support a causative role as opposed to collateral role for IFN in disease progression.

Response : We did not have the opportunity to follow IFN α level in HIV-infected patients from the onset of their infection.

However, comparisons of SIV infection in primate species that develop AIDS-like disease and species without disease symptoms indicate that an elevated IFN α serum concentration overflowing from lymphoid tissues to body fluid occur only during pathogenic infection in macaques, whereas natural SIV hosts, without disease progression, express residual background IFN α concentration levels (Mandl, J. N. et al. Divergent TLR7 and TLR9 signaling and type I interferon production distinguish pathogenic and nonpathogenic AIDS virus infections. *Nature Med.* 14, 1077–1087 (2008) ; Jacquelin, B. et al. Nonpathogenic SIV infection of African green monkeys induces a strong but rapidly controlled type I IFN response. *J. Clin. Invest.* 119, 3544–3555 (2009)). Similar findings have been made in individuals infected with HIV; rapid progressors show higher IFN α signatures than viremic non-progressors (Rotger, M. et al. Comparative transcriptomics of extreme phenotypes of human HIV-1 infection and SIV infection in sooty mangabey and rhesus macaque. *J. Clin. Invest.* 121, 2391–2400 (2011)).

It is noteworthy, as indicated in the revised companion paper, to point out that IFN α , like other cytokines produced in tissues of healthy individuals have a short half-life and express their message locally in a paracrine/juxtacrine manner on target cells expressing type I IFN α receptor. However, when abnormally produced, elevated level of IFN α , which overflow diluted into body fluid with concentration ranging from 10-1000 fg in UP serum.

Our data and these studies suggest that the initial dose of virus at infection drives the elevated level of IFN α and its pathogenesis: if the infection occurs in the presence of an elevated dose of virus, subsequently there will be a correlated elevated dose of IFN α , and a progression of IFN α pathogenesis leading to AIDS.

3: If, as the authors suggest, part of the mechanism is induction of CCR5 by IFN alpha and creation of more substrates for HIV replication, have they looked at the relative abundances of R5 versus X4 viruses in post- treatment interruption viremia?

Response : It is difficult to answer, considering that the rebound virus can originate from several cellular and anatomical compartments after treatment interruption (De Scheerder MA et al. HIV Rebound Is Predominantly Fueled by Genetically Identical Viral Expansions from Diverse Reservoirs. Cell Host Microbe. 2019 Sep 11;26(3):347-358.e7. doi: 10.1016/j.chom.2019.08.003. Epub 2019 Aug 27. PMID: 31471273)

On our observations so far, the R5 viruses are almost always predominant in circulating plasma. This could be mainly because R5 and X4 viruses are replicating in different CD4 subsets. While R5 viruses are mainly in effector memory and terminal TEMRA subsets compartments, whereas the X4 viruses preferentially target the central memory and naive subsets in lymphoid tissues. The different viral burst size (higher for the more activated/differentiated CD4⁺ T-cells where the R5 viruses replicate) provide R5 viruses advantage in body fluid (plasma).

In the revised paper, we added in Fig. 2 our experimental data showing that CCR5 but not CXCR4 expression in CD4⁺ T-cell cultured in presence of IFN α is enhanced in a dose effect manner (Fig. 2A3 and 2A4. We have also added the following sentence to the results section, at the end of the paragraph (penultimate line) entitled "High IFN α level enhances *in vitro* expression of the HIV coreceptor CCR5":

Interestingly, IFN α has no effect on CXCR4 expression at either mRNA level (Fig. 2A3) and protein level (Fig. 2A4) on stimulated CD4⁺ T-cells.

Reviewer #2 (Remarks to the Author):

Le Buanec and colleagues performed a comprehensive immunophenotyping with up-to-date bioinformatics analysis in untreated (UP, n=26) and treated HIV-infected patients (TP, n=21), uninfected donors (HD, n=24), and elite controllers (EC, n=12), focusing in this and the companion manuscript on interferon (IFN)- α -induced alterations.

Mains findings: The authors show significantly elevated peripheral IFN- α , but not IFN- λ 2 levels using SIMOA technology in UP compared to HD and TP. CCR5 mRNA and protein expression was upregulated in stimulated CD4⁺ T-cells upon exposure to increasing IFN- α concentrations. Pretreatment of stimulated CD4⁺ T cells (HD) with IFN- α reduced the inhibitory effect on a CCR5-, but not on a CXCR4-tropic virus. The count of NK cell subsets and their phenotype was altered in UP and not completely reversed in TP compared to HD. Similar results were obtained with distributions of CD4 and CD8 T-cell subsets and their expression of markers for activation and differentiation. In UP, Treg counts were increased, as were CTL and CD8+supp cells, which expressed increased markers of activation, differentiation and exhaustion. Finally, the authors show similarities in counts and phenotype of nearly all immune cell types between EC and TP.

The authors conclude that uncontrolled HIV-1 infection increases IFN- α secretion, which promotes further viral propagation via increased expression of coreceptor CCR5. Limiting this IFN- α secretion may contribute to a functional “cure” in HIV-1 infection. Early control of HIV-1 infection by cells of the innate immune defense, in particular NK cells, could reduce IFN-alpha production, contributing to elite controller status.

I do embrace the concept that elevated IFN- α levels play an important role in the HIV-1 immunopathogenesis. However, this concept is not new and has already been published by others (e.g. Jay Levy’s group), which should be cited (PMID 16278001, 22203858). Importantly, the authors show that phenotypic alterations that are present in UP and are partially reversed in TP, can be induced in vitro by IFN- α .

Response : We already answered this question in our response to the companion paper (please see this response).

The two manuscripts have not the goal to report that serum IFN α concentration is pathogenically elevated (observation already reported initially by us and confirmed by multiple reports as in those from Shearer’s and Levy ‘s groups. The present manuscript aimed at showing that besides HIV itself, elevated IFN α is a key mediator of HIV pathogenesis.

Nevertheless, to satisfy the referee's request, we have added, in addition to the Shearer's group papers, the paper by Levy's group in the revised accompanying document, both of which confirm our 1990' ones showing the pathogenicity of elevated IFN α in HIV infection.

Major points:

- The amount of peripheral IFN- α levels in UP ranges between 10-1000 fg/ml (Fig. 1D). The lowest IFN- α concentration that induces CCR5 mRNA upregulation was shown to be 10 pg/ml (Fig. 1A), while blocking of p24 secretion started with 0.2 ng/ml (Fig. 2C). Why are much higher IFN- α concentrations required for morphological changes in vitro? This is even more puzzling as UP have lower CCR5 on the CD4+ T-cell surface (at 0 ng/ml IFN- α) compared to HD, EC and TP (Fig. 2B). Following the authors' reasoning, shouldn't UP have the highest CCR5 levels at baseline?

Response : As other cytokines, IFN α has short half-life, is produced in tissues and particularly in lymphoid organs. IFN α acts on their target cells locally and is released into the body fluid at only background level (< 100 fg/ml in HD but also in most of EC and TP). However, when abnormally highly produced in lymph nodes, as in UP, IFN α abnormally overflows into the body fluid and becomes diluted in serum (at concentration of 10-1000 fg/mL). This serum concentration is obviously much lower than that present in tissues where they exert their action through specific high-affinity receptors (with Kd \approx 10–10 nM). The detection of the abnormal presence of IFN α in the serum can only be achieved with an ultrasensitive digital ELISA (Simoa).

During acute HIV infection, IFN α produced in lymph nodes, contributes to antiviral protection by induction of an ISG-based cellular antiviral program. The level of IFN α produced is correlated with the viral load. If the infection occurs in the presence of a high amounts of virus, subsequently there will be a production of an elevated level of IFN α , that is abnormally overflowing into body fluids is pathogenic.

The concentrations of IFN α used in CD4⁺ T-cells and NK cells culture correspond to that present in tissues, and not to that abnormally found in serum of UP. We would agree with the comment of referee 2 who considers it puzzling that UP have lower CCR5 in at the CD4⁺ T-cell surface, if we did not consider untreated patients with high viral loads whose viral particles bind to their CCR5 receptor resulting in internalization of this receptor.

- *Fig. 1A: is CXCR4 mRNA and protein expression also upregulated by IFN- α ?*

Response : When we measure the effect of IFN α on CXCR4 expression at protein and mRNA level, we do not observe an increase in CCR4 expression, but rather a slight decrease that varies according to the donor.

In the revised manuscript, we have added to the Fig. 2, new IFN α data on the expression of the two chemokine receptors CCR5 and CXCR4, confirming this claim (Fig. 2A3 and 2A4). We have also added the following sentence to the results section, at the end of the paragraph (penultimate line) entitled "High IFN α level enhances in vitro expression of the HIV coreceptor CCR5": Interestingly, IFN α has no effect on CXCR4 expression at either mRNA level (Fig. 2A3) and protein level (Fig. 2A4) on stimulated CD4⁺ T-cells.

- Fig. 2C: These data are key to the manuscript. No statistics was performed. The experiment should be repeated in another two to three donors and then the data of all donors should be summarized. Use different MOI to show the effect of IFN- α on the infection dose.

Response : Result shown in Fig.2C is typical of the 3-patient study. This is an illustrative experiment of the summary. Using different MOI to show the effect of IFN α on the dose of infection is interesting, but If MOI are high, of course, IFN α would not show visible effects, and of course too low and the experiment becomes impossible. We selected a doable MOI where infection can be seen, IFN α inhibitory effects may be noted (in CXCR4 HIV), and the decline in inhibition is evident in CCR5 HIV.

- *Statistics: which post-hoc test was used to control for multiple testing? What is meant by “biologically independent samples”?*

Response : Comparisons between groups were performed with unpaired nonparametric Mann-Whitney as indicated in the statistical analyses section of the material and methods of the manuscript.

In the revised manuscript, comparisons between two groups are performed with unpaired nonparametric Mann-Whitney and comparisons among multiple groups are performed using Kruskal–Wallis test followed by Dunn’s Post-hoc test. Methods of analysis were also indicated in the statistical analyses section of the material and methods in the revised text and in the figure legends.

Please accept our apologies for the lack of clarity for the sentence “biologically independent samples”? We simply wanted to say that each individual sample was analyzed once for each of the parameters studied.

So we remove “**biologically independent samples**” in the revised text.

- *How long were the TP actually treated at the time of sampling? It may make a difference for what period of time viral load was undetectable.*

Response : All TP received ART for at least 3 years and had HIV VL suppressed throughout this time.

Minor points:

- Methods, human samples: please provide reference numbers for ethical approval.

Response : The IRB number for the study in US is H-29331. In Liege (Belgium) the number of the agreement for the study is 2020/418. We added it in the material and method in the human samples section.

- Results, p. 5: The authors say that DC are increased in TP and UP compared to HD. Please specify whether PDC or MDC are meant here (also in Fig. 1B5). The percentage of PDC and MDC was calculated with respect to lin- HLA-DR+ cells (and not PBMC), which should be clearly indicated throughout the manuscript.

Response : In the Fig. 1B5, we present the frequency of DC in blood immune cells. However, indeed, the percentage of pDC and mDC was calculated with respect to lin- HLA-DR+ cells. We have now specified it in fig. 3B and in the legend as follows : **“Representative dot plot showing how to distinguish pDC (CD123+ CD11C-) and mDC (CD123-CD11C+) subsets within the HLA-DR+ lin- population in HD. Histograms showing the frequencies of pDC (B2) and mDC (B3) across the groups (HD n=22, UP n=10 and TP n=8)”**

- Discussion: The authors may want to speculate whether elevated IFN-alpha levels in UP cause the switch of CCR5- towards CXCR4-tropic viruses in the course of disease?

Response : We have no reason to speculate that high IFN α would change co-receptor although it is an interesting concept.

- Fig. 6: The authors show data of 12 EC, but the Extended data in Table 3 list 18 EC.

Response : To perform our phenotypic analysis, we had to prepare 2 cytometry panels. For some EC

patients, we lacked cells to perform the two panels, which is why in some figures there are 18 EC and in others 12.

- Abbreviations need to be better explained in the figure and table legends. Extended data Table 1: (med) means (median)? Extended data Table 3a: HIV dx? AA? Year of sample date – 012, 011? Table 3b: Excel sheet should be converted into a regular table. Figs. 6E1, 6F1, 6F4, 6F5: it is not clear what is meant by the scale. Is the cohort of UP or the highest value in all cohorts set as “1”?

Response : We removed the abbreviations in the titles of the legends and indicated in the legend what the abbreviation corresponds to when it appears the first time.

We modify the text as follows :

Legend Fig.1 : Title : Comparative analysis of major blood immune cell subsets and serum IFNs concentrations in Untreated Patients, Treated Patients and Healthy Donors.

Legend Fig.3 : NCR, GrzB/perf and iKIR were replaced by Natural cytotoxicity receptors, Granzyme/Perforin and Inhibitory Killer Ig-Like Receptors.

Concerning the Year of sample date, (– 012, 011), thank you for pointing out the typo. We fixed it in the revised manuscript text the date as follows: 2012, 2011

In extended table 1, we have added a legend to explain the abbreviations: NA : Not Available; med : median

In the extended tables describing the patients we have added a legend to explain the abbreviations: Dx : Diagnosis, M : Male, F : Female, AA : Afro American, IDU : Injection Drug Use, HS : Homosexual, MSM : Men who have Sex with Men, NA : Non Available.

As to the scale in Figs. 6E1, 6F1, 6F4, 6F5, the highest value in all cohorts is set as “1”.

- Both manuscripts access the same patients and partially overlap. On some occasions, data referred to in this manuscript (e.g. IFN- α levels in EC) are shown in the companion manuscript only. These parts need to be properly referenced.

Response : In order to facilitate the reading of the text, we have added in the 1st line of the 4th paragraph of the discussion chapter, the reference of the companion paper, reference 14, when we speak of the serum level of IFN α in ECs in the revised manuscript as follows :

“We show that EC and, to a lesser extent TP avoid these abnormalities of immune cell subsets (Figs. 2-6), and control IFN α levels and virus load (14) (Fig. 1 and Fig. 6A3)”

- The idea that elite controllers were infected by chance with a low viral load that does not induce ‘vicious IFN- α production’ is interesting but speculative. Conversely, EC might efficiently control HIV-1 infection in the short and (!) long run and therefore not respond with increased IFN-alpha production. To this end, it might be helpful to examine samples from seroconverters, which should be discussed.

Response : We of course agree with the reviewer on this point also questioned in the first manuscript. However, please keep in mind this is a hypothesis not meant to be solved here. The contribution of low infectious inoculum to the control of elevated IFN α is one hypothesis based on the following items:

- 1) Absence of the unique genetic parameter including HLA-B57, CCR5 Δ 32 and others;
- 2) Heterogeneous cytotoxic capacity of NK, CTL and CD8⁺ supp T-cells, variable between each EC observed in this study (text and Fig. 5 of the companion paper). We were struck by the fact that EC experts did not bring this up (the size of HIV inoculum) as far we know ever before because for us it seems the very likely answer that explains EC status.

3) Sequential analysis of sera from untreated HIV patients from the onset of infection which shows that a) IFN α is detected in serum after viral load detection, and b) serum IFN α levels correlate with viral load

levels (Hardy GA et al. Interferon- α is the primary plasma type-I IFN in HIV-1 infection and correlates with immune activation and disease markers. *PLoS One*. 2013;8(2):e56527. doi: 10.1371/journal.pone.0056527. Epub 2013 Feb 20. PMID: 23437155; PMCID: PMC3577907; Stacey AR et al. Induction of a striking systemic cytokine cascade prior to peak viremia in acute human immunodeficiency virus type 1 infection, in contrast to more modest and delayed responses in acute hepatitis B and C virus infections. *J Virol*. 2009 Apr;83(8):3719-33. doi: 10.1128/JVI.01844-08. Epub 2009 Jan 28. PMID: 19176632; PMCID: PMC2663284.)

4) It is well known that patients with high viral load “set points” experience disease progression much more rapidly than those with lower viral load set points. Elite controllers, therefore, represent the extreme of those with low viral load set points, corresponding to correlated low IFN α production, sufficiently controlled during the acute infection by innate immune defense mechanisms (IFN α , IFN λ and NK-cells) and during adaptive phase by additional cytotoxic cells. (Goujard C et al. Spontaneous control of viral replication during primary HIV infection: when is “HIV controller” status established? *Clin Infect Dis*. 2009 Sep 15;49(6):982-6. doi: 10.1086/605504. PMID: 19681706; Okulicz JF et al. Infectious Disease Clinical Research Program (IDCRP) HIV Working Group. Clinical outcomes of elite controllers, viremic controllers, and long-term nonprogressors in the US Department of Defense HIV natural history study. *J Infect Dis*. 2009 Dec 1;200(11):1714-23. doi: 10.1086/646609. PMID: 19852669; Woldemeskel BA et al. *EBioMedicine*. 2020 Dec;62:103118. PMID: 33181459; PMCID: PMC7658501)

5) Some Elite controller lost their Elite controller status especially due to an episode of viral superinfection associated with high IFN α production. We have seen that two elite controllers in our cohort lost their status (EC11 and EC52) following HCV infection associated with an increase in antiviral IFN α . A study published by Rossol S et al also shows, that a high presence of high levels of serum IFN α , notably, preceded the loss of virological control in elite controller (Rossol S et al Interferon production in patients infected with HIV-1. *J Infect Dis*. 1989 May;159(5):815-21. doi: 10.1093/infdis/159.5.815. PMID: 2468718).

All of this is now discussed in the revised manuscript.